

# Variability of depolarization of aerosol particles in Beijing mega city: implication in interaction between anthropogenic pollutants and mineral dust particles

Yu Tian[1, 2], Xiaole Pan[1, *], Tomoaki Nishizawa[3], Hiroshi Kobayashi[4], Itsushi Uno[5], Xiquan Wang[1], and Zifa Wang[1,2,6]

[1]State Key Laboratory of Atmospheric Boundary Layer Physics and Atmospheric Chemistry, Institute of Atmospheric Physics, Chinese Academy of Sciences, Beijing, 100029, China
[2]University of Chinese Academy of Sciences, Beijing, 100049, China
[3]National Institute for Environmental Studies, Tsukuba, Ibaraki, 305– 8506, Japan.
[4]University of Yamanashi, Yamanashi, 400–0016, Japan.
[5]Research Institute for Applied Mechanics, Kyushu University, Kasuga, Fukuoka, 816–8580, Japan.
[6]Center for Excellence in Regional Atmospheric Environment, Chinese Academy of Science, Xiamen, 361021, China

*Correspondence to*: Xiaole PAN (panxiaole@mail.iap.ac.cn)

**Abstract.** East Asia is suffering from a severe air pollution problem due to intensive emissions of anthropogenic pollutants and mineral dust aerosols. During transport, both Asian dust and pollutants undergo complex mixing processes, which result in great impacts on regional air quality, human health and climate. To characterize these mixing processes, we conducted long-term observations using an optical particle counter equipped with a polarization detection module (POPC) in the megacity of Beijing. Mass concentrations of $PM_{2.5}$ (aerodynamic diameters of less than 2.5 µm) and $PM_{10}$ derived from POPC compared well with ground-based measurements. Based on size-resolved polarization measurements, anthropogenic pollutants, mineral dust and mixed-type mineral dust (dust particle with pollutant coatings) were well distinguished. We found that the depolarization ratio (DR, termed as the ratio of the intensity of the s-polarized signal to the intensity of the backward scattering signal [s/(s + p)]) for aerosol particles with optical diameters of < 2.0 µm has a pronounced seasonal variation trend, with a winter-spring high and a summer low pattern. Aerosols that originated from Mongolia and western China normally demonstrate non-spherical shapes, with a DR centred on 0.3, whereas the value for fine-mode particles was usually less than 0.1 due to the predominance of secondary water-soluble matter. In Beijing, the morphology of mineral dust changed obviously, particularly in humid conditions. Higher relative humidity accelerated the liquid phase reaction and promoted water absorption due to the coating of hygroscopic soluble compounds on the dust surface. This study also found that 48% of substandard air quality days featured high coarse-mode particles in the winter-spring time, which implies the great influence of heterogeneous processes on the surface of mineral dust on smog formation.



# 1 Introduction

Rapid economic growth in East Asia has caused serious air pollution in the past decades owing to the substantial consumption of fossil fuel, which emits large amounts of pollutant gases (SO$_2$, NO, NH$_3$, VOCs, black carbon, etc.) (Akimoto, 2003; Akimoto et al., 2006; Kurokawa et al., 2013). Based on a former study, the total energy consumption in Asia more than doubled in 2003 compared to 1980 (Ohara et al., 2007), and this severe emission situation will continue (Cofala et al., 2007). Mineral dust particles from natural sources also have a detrimental impact on air quality at their downwind location. For example, Beijing (capital of China) is located just hundreds of kilometers from the Taklamakan Desert in western China and the Gobi Desert, on the border between south Mongolia and north China. In dry seasons, Asia's bare grasslands, arable land, desert and the Gobi all become dust sources, and Beijing is located just in the dust plume's eastward transport path (Takemura et al., 2002; Jickells et al., 2005; Uno et al., 2009). Tens of thousands of tons of mineral dust enter the atmosphere due to strong winds and are transported eastward across the northern part of China, resulting in a mixture of anthropogenic pollutants (e.g., sulfate, nitrate, ammonium salt) and mineral dust (Li and Shao, 2009; Nie et al., 2015).

The environmental and climate effects of these complicated mixing processes are notable because of dramatic changes in the physical, chemical and optical properties of mixed particles (He et al., 2014; Pan et al., 2009). In polluted urban area, soluble salts coated on dust aerosols reduce the critical supersaturation, and there is a stronger tendency for polluted aerosols to serve as CCN (cloud condensation nuclei), which promote the indirect effects of dust aerosols (Tang et al., 2016; Sullivan et al., 2009). The soluble salts are derived from directly trapping sulfate, nitrate, etc., as well as the heterogeneous reactions between reactive gases (e.g., HNO$_3$, HCl, SO$_2$ and NO$_2$) and alkaline mineral dust (Krueger et al., 2004; Laskin et al., 2005). Continuous coating and hygroscopic growth processes on the surface modify the shapes of dust particles. A recent study pointed out that coexistence of NO$_X$ and mineral dust may lead to a gas-particle conversion process and promote the conversion of SO$_2$ to sulfate (He et al., 2014). There is also observational evidence that heavy dust mixed with anthropogenic pollution may trigger new particle formation, which exaggerates the degradation of regional air quality (Nie et al., 2015). The above scientific discoveries all indicate the importance of studying the mixing states of dust and pollution aerosols.

Widely used technologies to detect aerosols include high-precision ground-based LIDAR (Light Detection And Ranging) systems and satellite-borne observations (Cloud-Aerosol Lidar and Infrared Pathfinder Satellite Observation; CALIPSO), which are instruments that measure an air parcel's volume depolarization ratio ($\delta_a$: ratio of the volume of perpendicular and parallel backscatter intensity, i.e., S-polarized in contrast to P-polarized in reflected component) to distinguish different types of aerosols (Burton et al., 2012; Winker et al., 2009; Shimizu et al., 2004; Kim et al., 2004). The merit point is that aerosol types can be investigated by the distinct parts formed by data points on a figure of their backscattering coefficient ratio (1064 nm/532 nm) versus $\delta_a$ (at 532 nm); however, bias in the classification of mixed types is unavoidable since external mixing of a substantial amount of fine particles ($\delta_a < 0.1$) with mineral dust aerosols ($\delta_a > 0.35$) also results in a lower $\delta_a$, but the measurement of $\delta_a$ on a single-particle basis could accurately characterize such an interaction. To respond to this need, an



optical particle counter with a depolarization module was developed. Until now, a number of studies in East Asia have been conducted that focus on the polarization properties of single particles (Pan et al., 2016; Sugimoto et al., 2015; Kobayashi et al., 2014). However, measurements in a megacity in North China where the ambient mass concentrations of both anthropogenic pollution and mineral dust were high are still lacking. Studies on the polarization characteristics of single particles in a Chinese megacity are thus rare and necessary.

In this study, aerosols' morphological features were continuously measured using a polarization optical particle counter. Dust, anthropogenic pollutants and polluted dust were classified according to their polarization properties. The instrument POPC was installed on the second story of a two-story building located in the northwest corner of the State Key Laboratory of Atmospheric Boundary Layer Physics and Atmospheric Chemistry (LAPC, Longitude: 116.37E; Latitude: 39.97N), Institute of Atmospheric Physics/Chinese Academy of Sciences in Beijing (shown as the red circle in Figure 1). Key pollutants, including the mass concentrations of $PM_{2.5}$ and $PM_{10}$ came from the Olympic Sport Centre state control station (Longitude: 116.40E; Latitude: 39.98N). Meteorological data were obtained from the Beijing Chaoyang district meteorological station (116.48E, 39.95N), and include meteorological factors such as wind (degree & direction), relative humidity, temperature, etc. The sphericity of aerosols from different sources, the mixing state and its dependence on the environmental water content were also investigated. The air mass's long-range transport was calculated using the FLEXPART Dispersion Model and an ensemble simulation of multiple trajectories using the Hysplit Trajectory Model.

This study focuses on the seasonal variation of the depolarization ratio of aerosol particles and its relationship with secondary pollutants. This represents the first time that the long-term variability of ambient aerosol morphology in the megacity of Beijing has been studied based on data of single particle polarization properties, and this study has high scientific value for understanding the mixing processes of atmospheric aerosols and their climate impact.

## 2 Observation

### 2.1 Observation Instruments

Continuous observation of the polarization properties of single particles in Beijing was made using the Polarization Optical Particle Counter (POPC). (Kobayashi et al., 2014). Figure 2 shows the schematic diagram of the POPC. When the ambient air mass enters the POPC under the suction of the pressure pump, the chaotic particles are arranged in a mono-dispersive aerosol beam under the action of sheath zero gas and pass through the measurement chamber in line, and a polarized laser beam at a wavelength of 780 nm illuminates the particles. Forward scattering light is detected at a scattering angle of 60° to determine the size and number concentration of particles, and the backward scattering intensity is detected at 120°. Backscattering light was made up of two components; P-polarized light is in the plane of the incident and reflected beams, known as the horizontal polarized light, while S-polarized light is perpendicular to the plane. Normally, the ratio of the S-polarized signal to that of P is defined as the depolarization ratio (i.e., DR value) (Shimizu et al., 2004), which can provide morphological information about the particle (Muñoz and Hovenier et al., 2011). The acceptance angle (angle of the



backscattered light received by the polarizer) for the polarization detector is 45°. For non-spherical particles, the vibration of the 120° scattered light is asymmetric in the propagation direction (polarized), which is indicative of a relatively large DR value. To avoid coincidence error (several particles passing through the laser beam simultaneously) of the instrument at conditions of a high number concentration, the aerosol flow rate is set to 80 LPM (Liters Per Minute) and diluted with

5 absolute clean air at 920 LPM. Combined with pollution concentration data (including $PM_{2.5}$, $PM_{10}$, $SO_2$, $NO_2$, CO, etc.) from the ground state control site, and corresponding meteorology data from the climatological station, we analyzed the artificial pollution process and special mineral dust cases that occurred in Beijing.

## 2.2 Footprint and trajectory analysis

Footprint simulations use the FLEXPART Dispersion Model. FLEXPART (FLEXible PARTicle Dispersion Model) is a

10 Lagrangian transport and dispersion model (https://www.flexpart.eu) developed by NILU (Norwegian Institute for Air Research). This model is suitable for the simulation of a large range of atmospheric transport processes (Stohl et al., 2005). The meteorological fields for FLEXPART are taken from NCEP's (National Centers for Environmental Prediction) reanalysis GDAS (Global Data Assimilation System) dataset on a 1° lat × 1° lon grid, which provides global observation meteorological data at 00, 06, 12 and 18 UTC and forecast data at 03, 09, 15, and 21 UTC

(http://nomads.ncep.noaa.gov/pub/data/nccf/com/gfs/prod/). During the simulation, 1 unit mass of particles considered as an air sample was released from the observation site at 1000 m above ground level. The spatial distribution of the footprint region of the air samples was calculated on the 5 days of backward simulation considering flow meteorology, turbulent motions, the subgrid terrain effect and Earth's water cycle.

A multitime collection footprint simulation used the HYSPLIT (Hybrid Single Particle Lagrangian Integrated Trajectory

Model) Trajectory Model developed by NECP and NCAR (National Center for Atmospheric Research), which is based on the Lagrangian transport model (https://ready.arl.noaa.gov/HYSPLIT_traj.php) (Stein et al., 2015). The dataset provided for HYSPLIT is the global reanalysis data in GDAS format that was stored for 5 weeks per month (ftp://arlftp.arlhq.noaa.gov/pub/archives/gdas1). It produces meteorological data four times a day, namely, at 00, 06, 12, 18 (UTC), and the horizontal resolution is 2.5° × 2.5°. The vertical direction is 17 floors, ranging from the ground surface to 10

25 hPa. Elements, including wind, temperature, humidity, potential height and ground precipitation, are provided. During the simulation, the trajectory ensemble option starts multiple trajectories from the first selected starting location. Each member of the trajectory ensemble is calculated by offsetting the meteorological data by a fixed grid factor (one grid meteorological grid point in the horizontal and 0.01 sigma units in the vertical). Air samples were released at 150 m above ground level, and the simulation time of the backward trajectory is 5 days.



## 3 Results and discussion

### 3.1 Size distribution of ambient aerosols

The hourly particle volume size distribution from November 29, 2015 to July 29, 2016 is shown in Figure 4a. The particle size was derived according to the calibration curve between the forward scattering intensity and spherical standard particles' diameter data (Supplementary Figure S1). To test the accuracy of the POPC detection, the $PM_{2.5}$ and $PM_{10}$ inverted by POPC were compared with the observed data from the Olympic Sport Center state control station, as shown in Figure 3. The correlation coefficients between the POPC and state control station are 0.91 and 0.86 (significance level: 0.001) for $PM_{2.5}$ and $PM_{10}$, respectively, with a sample size of more than 1000 data, which proved the credibility of the POPC measurements. As a general rule, there were two dominant particle size ranges in the ambient aerosols. The fine mode (<1 μm) was mostly ascribed to anthropogenic pollution, while the coarse mode (4 μm – 8 μm) can be explained by sandstorms during strong wind and dry conditions in the spring (Kurosaki and Mikami, 2003). The growth of fine mode particles mainly stems from high anthropogenic emissions during heating seasons. The distinct growth period of fine mode particles was usually induced by an increase in relative humidity (i.e., RH), for example, cases on Nov. 23[rd], Mar. 3[rd], Mar. 15[th], and Jun. 6[th], etc., shown in Figure 4a. High ambient humidity can promote non-homogeneous reactions, accelerate transformations between gas and particles, and cause the corresponding fine mode aerosols' concentrations to rise. The effect of the environmental water content is discussed in part 3.5. Considering the variation of both the volume size distribution and depolarization ratio, select anthropogenic pollution, mixed type pollution and pure Asian dust cases from Figure 4 are noted as "A" (December 22[nd] to 23[rd], 2015), "B" (March 3[rd] to 4[th], 2016) and "C" (April 9[th], 2016), respectively.

### 3.2 Depolarization ratio of ambient aerosols

Figure 4b illustrates the size-resolved depolarization ratio as a function of time during observations in Beijing. The depolarization ratio rises significantly as the particle size increases, since there is a larger S component in backward scattering and the huge irregularity of coarse mode particles. Episodes influenced by mineral dust could be easily discerned due to the increase in both the volume concentration of coarse mode aerosols and the DR value of fine mode aerosols (e.g., Case C). In reviewing a longer period of time, the depolarization ratio of fine mode particles has prominent seasonal variabilities with a summer-low and spring-high pattern. In contrast, in the summer, particles with an optical diameter of 2 μm (with a specific DR value of 0.2) have an obviously lower depolarization ratio (near 0.1 in Jul.) than any other season.

In order to understand the major orientation of air masses in different pollution periods and explain the seasonal characteristics of the depolarization ratio of atmospheric aerosols, the backward footprint of the air mass in severe anthropogenic pollution cases and mineral dust dominant processes were analyzed. Specific pollution incidents were chosen based on the size-resolved volume distribution and depolarization ratio (**Figure 4**). Practical criteria were: Air Quality Index (AQI) > 300, $PM_{2.5}$ > 250 μg/m$^3$ and $PM_{2.5}/PM_{10}$ > 0.70 (except on Nov. 30[th], 2015) for heavy anthropogenic contamination days; $PM_{10}$ > 150 μg/m$^3$ and $PM_{2.5}/PM_{10}$ ≤ 0.40 (except on Mar. 31[st], 2016) for dust-dominated cases. See Table 1 for





concrete details of the selected simulation days. The backward trajectory was derived from HYSPLIT ensemble calculations, resulting in 27 members for all-possible offsets around the release point. The trajectory of the air mass at each moment was simulated for the previous 120 hours, and the proportions of different directions from which the air mass originated to Beijing are shown in **Figure 5**. For the days when coarse mode aerosols dominated, the air mass mainly originated from

Mongolia in the northwest, transnational Inner Mongolia with the Gobi Desert being the main transit area. This pattern explains why the monthly averaged depolarization ratio of all particle size ranges in Beijing can reach 0.26 in March, which is the highest over the observation period. However, for days on which fine particles dominated, a certain proportion of air masses still came from the northwest region for Beijing's location in the control zone of the Westley belt, and a notable major source orientation of air masses was southwest of Beijing, blowing huge amounts of pollutants from the

manufacturing district (right panel of **Figure 5**).

The seasonal periodic fluctuations of the depolarization ratio can also be seen in Figure 4c. It is a time series of hourly and monthly averaged DR for the typical particle size. For fine mode particles (1 μm), the monthly averaged DR value was apparently higher in the spring, with the highest peak in April (~0.13). This spring high feature can be ascribed to the fine mode part in the mass distribution spectrum of dust aerosols. In contrast, the seasonal change of coarse mode particles (5 μm)

shows no heaving, which means that the change in morphology of dust particles in different seasons was less evident, which partially explains the weaker reactiveness of mineral dust.

### 3.3 Air quality characteristics in the Beijing area

The box chart of hourly mass concentrations of $PM_{2.5}$ and $PM_{10}$ per month measured at the Olympic state control site is shown in Figure 6. In the timeline view, $PM_{2.5}$ showed a certain consistent trend with $PM_{10}$, in which the maximum value

occurred in December, 2015 (monthly averaged $PM_{2.5}$: 157.46 μg/m³, and $PM_{10}$: 199.60 μg/m³), and a secondary peak appeared in the spring of Mar. and Apr. (i.e., MA). However, $PM_{2.5}$ only accounts for 62% of $PM_{10}$ in MA, while this value reached 0.79% in the winter (i.e., Dec. and Jan.; DJ). In DJ, the median value of $PM_{2.5}$ (Dec.: 115.83 μg/m³; Jan.: 44 μg/m³) approached the lower quartile (25th percentile) (Dec.: 48.10 μg/m³; Jan.: 12.83 μg/m³), along with a high upper limit (90th percentile) (Dec.: 322.542 μg/m³; Jan.: 142.71 μg/m³), which indicate the occurrence of intensive anthropogenic pollution

incidents.

Figure 6c shows the number of poor air quality days per month. Here, the date when the daily-averaged mass concentration of $PM_{2.5}$ exceeded the secondary standard of the Chinese Ambient Air Quality Standard of $PM_{2.5}$ (75 μg/m³) is defined as a substandard day, i.e., a poor air quality day. As seen in Figure 6c, the numbers of polluted days in Dec. (17 days) and Mar. (15 days) were the greatest during the observation period, and 81% and 80% can be attributed to mixed type pollution corresponding to the two months (Figure 6d). Mixed type pollution was also necessary for substandard days in Jan. (44%),

Feb. (50%), and Apr. (35%). In the summer (JJ), however, the contribution of mixed pollution was muted, i.e., 8%. To summarize, the deterioration of air quality in the megacity in north China was closely correlated with the mix of dust and local emissions, especially during the winter-spring time. A potential reason was that the catalyzing process on the surface of



mineral dust aerosols affects the formation and evolution of haze. A key driver of atmospheric nucleation was from the production of $H_2SO_4$, which was generated from the oxidation of $SO_2$ (Kulmala et al., 2006), and previous research found that mineral dust coexisting with NOx can promote the conversion of $SO_2$ to sulfate (He et al., 2014). In addition, Nie et al. (2014) found that mixed plumes provide abundant reactive species (e.g., OH, $NO_3$, HONO, NOx), and dust-induced

photocatalytic reactions accelerate oxidation in $SO_2$ and volatile organic compounds (VOCs), and lead to new particle formation. Also, the photo-oxidation of VOCs reduces the volatility of organic matter, and the condensation process of these low-volatile vapors also affects fog formation. On these grounds, this kind of polluted dust induces heterogeneous photochemical reactions and is a major factor in reducing Beijing's air quality, especially in the winter and spring, confirming the observational data.

### 3.4 Polarization properties of polluted dust aerosols

Li et al. (2011) showed that mineral dust aerosols can be involved in heterogeneous reactions with gaseous pollutants, leading to changes in dust size, shape and chemical components. In addition, the optical and hygroscopic properties of dust particles changed as a consequence, which can directly and indirectly affect climate effects. Since the morphology of aerosols varies substantially, it depends on the mixing state while suspended in the air or transported by wind flow, but it can

be described by its polarization properties. Three pollution incidents were picked, as mentioned in section 3.1, to analyse the size and polarization properties of different types of aerosols. The backward trajectories from Beijing calculated by the FLEXPART dispersion model for the three cases are shown in Figure 7a, b and c, and the surface plot of depolarization ratio with respect to particle size are depicted in Figure 7d, e and f. The standard volume concentration is used to represent the volume distribution, which is calculated using the normalizing formula: Standard Volume Concentration = (truth value –

smallest value) / (largest value - smallest value). Figure 8 shows the synchronized crucial meteorological conditions and air quality index values.

The optical properties of particle swarms were markedly different, as Figure 7 shows. In case A, the air mass was mainly transported from the southwest of Hebei Province, passing cities like Shijiazhuang, Langfang, and Baoding, which are all areas with high emissions (Figure 7a). From the 21st (Dec.) to the 23rd (Dec), the ground wind was weak (below 1.5 m/s);

however, the ambient air humidity was high and was approximately 90% after the 22nd (Figure 8). Weak diffusion conditions and high relative humidity led to the explosive growth of $PM_{2.5}$ levels beginning from the 22nd (Dec.), at 12 o'clock. The volume concentration of particles in this period was concentrated in the submicron scale, with a depolarization ratio of less than 0.1 (Figure 7d), and this type of distribution was mostly supposed to be secondary formation pollutants. In case B, the air mass was a convergence of deviating northwest and south streams. The eastward-conveying air mass originated from the

Taklimakan Desert in Sinkiang and channeled the Tengger Desert in Inner Mongolia while flowing east; the south branch of the air mass was from central Henan Province and passed through southern Hebei Province in the process of moving north (Figure 7b). Famous mining areas in China, including Jiaozuo, Zhengzhou and Hebei, were located on the latter transmission path, resulting in a huge amount of dust that converged with the pollutant-rich gaseous atmosphere in the Beijing area.



During this period, the ground wind degree was 0-2 (i.e., 0-1.5 m/s) of the observatory, and the relative humidity ranged from 50% to 90%, providing good external conditions for the inner mixing of mineral dust aerosols and pollutants. However, as for the polarization properties, the external mixing characteristics were still obvious (Figure 7e): centered on the lower-left part represents anthropogenic aerosols with small diameters (<1 μm) and DR values (<0.1), while another maximum region

shows properties of dust particles with large diameters (2 μm) and DR values centered on 0.3. This two-peak pattern is determined by dominant mechanisms and meteorological conditions. Itahashi et al. (2010) found that two consecutive low-pressure systems over the observation site will lead to a dust layer lifted up within the warm-sector of the low-pressure system, resulting in unmixed dust and pollutants. However, if the observation site was behind a cold front, the pollutants will be trapped by dust and the two will mix well with each other within the PBL, forming polluted dust.

The percentage of $PM_{2.5}$ in $PM_{10}$ for the first processes (22$^{st}$ -23$^{rd}$, Dec. 2015) was relatively high, at approximately 80%. In contrast, in the third process (9$^{th}$, Apr. 2016), $PM_{2.5}$ only accounted for 0.22% of $PM_{10}$. The results of the trace analysis for case C indicate the air mass originated from Mongolia (Figure 7c) with a greater wind degree at level 1-3 (0.3 m/s – 5.4 m/s) compared to the previous two cases, accompanied by humidity of less than 40% for the whole time. Obviously, the meteorological conditions were not conducive to non-homogeneous reactions between alkaline dusts and artificial reactive

gases, and primary anthropogenic pollution emissions in this period were also lower compared to the winter time, because the heating season was over (normally from November 15th to March 15th of the next year in northern China). Above all, this case can certainly be defined as a pure dust process. The variations in the depolarization ratio as a function of particle size depicted in Figure 7f show the polarizing characteristics of diameters larger than 3 μm, with DR values that varied from 0.1 to 0.5. The wide range of DR values can be attributed to the dust particles' irregular shapes and the variable angles at

which the laser beam hit the particles. To summarize, the POPC instrument can help identify anthropogenic and mineral dust aerosols, and the crucial value is believed to be 0.1 in the megacity of Beijing to separate non-spherical particles. Notice that this key value can only be used for the analysis of POPC measurements.

The time-averaged volume size distributions and corresponding mode distributions of the depolarization ratio in the above study cases are depicted in Figure 9. For the volume distribution, the peak diameter of heavy anthropogenic pollution was

25 0.9 μm, shown as a blue line in Figure 9a, in contrast with case C, in which maximum value lay in the coarse mode size range (5 μm). Focus on case B, where there were two volume concentration peaks corresponding to the particle sizes at 0.9 μm and 4.5 μm in Figure 7e. This bimodal distribution was a canonical distribution for the external mixture of mineral dust aerosols and pollutants. As mentioned, in this period, the wind speed was between 0 and 1.5 m/s, and the relative humidity was between 50% and 90%; a credible speculation based on these relatively stable and humid conditions was collision

contact, and internal mixing may happen under the coexistence of two kinds of aerosols. Based on the depolarization ratio size distributions in cases B and C (right panel in Figure 9b), we found that the DR values of particles larger than 1 μm in diameter were lower in the mixed pollution case than in the pure dust case. This obvious distinction implied the possibility of alteration in the complex refractive index (the real and imaginary part) or morphology changes in coarse mode aerosols, i.e., dust particles. The depolarization ratio at a 120 degree scattering angle (consistent with the POPC detection angle of



backward scattering) decreased as the imaginary part of the refractive index increased (Ishimoto et al., 2010; Dubovik et al., 2006). However, Pan et al. (2017) reported that variations in the refractive index of a particle can only explain limited depolarization variability (5%) on the basis of the T-matrix methodology (Dubovik et al., 2002). In other words, the observed change in the depolarization ratio in coarse mode particles in case C was mainly caused by morphological changes.

With the indication of the occurrence of chemical reactions and coating processes on the surface of mineral dusts aerosols, the effects and principal influence factors of the mixing degree will be discussed in part 3.5.

**3.5 Synergistic influence of meteorology and mixing processes on the DR value**

Polluted air in Beijing contains high levels of reactive gases, including $HNO_3$, $SO_2$, $NO_2$ and $HCl$, as well as secondary sulfates. Both coagulation processes of ambient soluble matters and heterogeneous reactions of mineral dust and reactive

gases accelerate the formation of the surface crust on dust aerosols. There was observational evidence of soluble salts ($Ca(NO_3)_2$) coated on mineral dust in pollution days in northern China (Li and Shao, 2009). According to Sullivan et al. (2009), pure $CaCO_3$ and $CaSO_4$, which have diameters of approximately 2 μm, need supersaturation values of 0.6 to 0.9, respectively, to be activated. In contrast, the soluble salts of $Ca(NO_3)_2$ and $CaCl_2$, are more likely to be activated with supersaturation values ranging from 0.07 to 0.4. This means that the more dust particles become involved in chemical

mixing, the more easily they become hydrophilic and are incorporated into cloud processes (Koehler et al., 2009; Tang et al., 2016). Here, the vital roles that ambient air humidity and mixing degree played in the morphological changes of particles were investigated on the basis of the DR value as a function of relative humidity and the proportion of $PM_{2.5}$ in $PM_{10}$ (indicator of the mixing degree).

Figure 10 shows the surface plot of the depolarization ratio as a function of particle size and the ratio of $PM_{2.5}/PM_{10}$. Again,

the increase in particle size played a vital role in the increase of the depolarization ratio. A distinguishing feature was that the hourly averaged depolarization ratio of the particles decreased as the ratio of $PM_{2.5}/PM_{10}$ increased. For fine mode particles, e.g., 1 μm, the DR values range from 0.06 to 0.14 as $PM_{2.5}/PM_{10}$ decreased from 1 to 0.2. For coarse mode particles (take 5 μm for example), the value rose from 0.24 to 0.3 as the proportion of $PM_{2.5}$ in $PM_{10}$ decreased by the same amount. One of the reasons for this was that $PM_{2.5}$ was mainly composed of anthropogenic pollutants and only a small amount of mineral

dust matter. Another possible reason was the high concentration of $PM_{2.5}$ reflects high emissions to some extent, implying affluent reactive gases in the air. Previous studies confirmed that dust particles can uptake anthropogenic reactive gases and form secondary coatings after they are transported over an area with high anthropogenic emissions (Nie et al., 2015; Liu et al., 2008). The deeper the mixing state was, the greater the coating process and the bigger the morphology change were. This kind of surface change can be described by the depolarization ratio. In addition, in a polluted dust plume, the reaction of

alkaline $CaCO_3$ with gaseous $HCl$ and $HNO_3$ led to more hygroscopic soluble salts (compared to other calcium salts, $CaSO_4$, e.g., the production of $CaCO_3$ and $CaSO_4$) coated on the surface of dust particles. The correlated simulation of heterogeneous



reactions and its importance between artificial pollution and mineral dust aerosols has also been made and confirmed (Wang et al., 2017).

Actually, the morphology change of coarse mode particles is highly affected by metrological conditions, especially the effect of vapor content in the air, as **Figure 11** suggests. When the ambient relative humidity increased from 10% to 100%, there

was a noticeable increase of $PM_{2.5}/PM_{10}$ (0.15 to 0.9) due to the accelerated formation of secondary pollution aerosols by liquid phase chemical reactions (**Figure 11** c & d). With the increase of ambient humidity, the depolarization ratio of coarse particles also showed a decrease. For example, with 3-μm particles, the DR value can reach 0.3 in dry air conditions (RH was 10%) and be as low as 0.15 when the humidity approaches 100%. This kind of decrease in the depolarization ratio reflects the diminishing of the dust particles' aspect ratio (the ratio of the macro axis to the minor axis of an ellipsoid), which

was apparently explained by continuous heterogeneous reactions and coagulation processes in the humid air. Kelly and Wexler (2005) found that crystalline hydrates of $Ca(NO_3)_2$ and $CaCl_2$ formed at 9% and 0.6% RH as the critical values, and the corresponding deliquesce points were 50% and 28% RH, respectively. In mid-latitude regions, the typical ambient relative humidity ranges over 10% – 90%; thus, typical calcium salts in the atmosphere, such as $Ca(NO_3)_2$ and $CaCl_2$, coating the outer layers of mineral dust aerosols are generally in the dissolution state. In a humid environment, alkalescent

dust in the air is more susceptible to the attachment of water vapor, promoting the deliquescence of soluble substances on the surface. Water film also acted as a solute on the dust particles' surface, and the rate of the liquid phase reaction is much higher compared to a gas reaction. Pan et al. (2017) demonstrated that the increase of the water vapor content contributed to the decrease of the depolarization ratio of the dust particles in the presence of the $Ca(NO_3)_2$ coating. To sum up, after the internal mixing of mineral dust aerosols and anthropogenic pollutants, the hygroscopicity of dust particles increased. As the

atmospheric moisture increases, the water vapor-imbibing and $Ca(NO_3)_2/CaCl_2$ deliquesce process occurs on the surface of dust particles. However, all these conclusions were just deduced from this observation experiment, and more powerful evidence still needs to be tested in the lab.

## 4 Conclusions

In this study, we conducted continuous observations of atmospheric aerosols in the megacity of Beijing from November 29,

2015 to July 6, 2016 using a Polarization Optical Particle Counter (POPC). Combined with a backward trajectory analysis, the features of pure Asian dust, polluted mineral dust and locally emitted anthropogenic pollutants were investigated. The findings can be summarized as follows:

The depolarization ratio ($\delta_a$) of particles in urban Beijing has significant seasonal variabilities due to different components and origins of aerosols. The averaged depolarization ratio of all mode particles in the air was highest in Mar. 2015 (0.26) and

30 lowest in Jul. 2016 (0.19). The seasonal variation of the depolarization ratio in fine mode particles (<2 μm) was more obvious and showed a typical winter-spring high and summer low pattern. This observed seasonal fluctuation is related to primary emissions as well as weather conditions in different seasons. In addition, the mixture of mineral dust aerosols and locally emitted pollutants played a vital role in poor air quality days in Beijing, especially during the winter-spring time. The



specific contributions of the mixing effect to total substandard days were 81% and 80% in Dec. 2015 and Mar. 2016, respectively.

The POPC can help identify the sphericity and mixing state of ambient aerosols based on their depolarization properties. Unlike LIDAR and satellite-borne detectors, the instrument obtains information by testing single particles. The critical value of the depolarization ratio is believed to be 0.1 for the sphericity of particles in Beijing. Notably, this crucial $\delta_a$ obtained by POPC is different from those observed by LIDAR observations, and the direct application of LIDAR data to aerosol classification is problematic.

During severe haze and sandstorm days, the surfaces of aerosol particles may be changed because of the trapping process of soluble salts and heterogeneous reactions with reactive gases. This internal mixing process is largely influenced by meteorological conditions. With sufficient water vapor and low wind velocity, alkaline dust suspended in the air has more of a chance to touch moisture and contaminants, which will speed up liquid phase reactions and the deliquescence of soluble matter on the surface, finally leading to the accelerations of the coating process.

These findings bring attention to a deeper understanding of the mixing characteristics and possible mechanisms of aerosols' morphological change in East Asia. In addition to LIDAR and satellites, a reliable detection method for separating aerosol types and investigating mixed states is provided from the perspective of single particle polarization properties. This study also revealed the significant contribution of mixed pollution to substandard days in urban cities, indicating an urgent need for a module in a numerical model for the reactions involving dust particles during transport.

## Acknowledgement

This work was supported by the National Natural Science Foundation of China (Grant No. 41675128 and 41620104008). Air quality data was provided by ministry of environmental protection, and meteorological data was from the Chinese meteorological department. The author also thank the NCEP for provision of GDAS dataset, NILU for develop FLEXPART Transport and Dispersion model, and NCAR for exploiting HYSPLIT Trajectory Model collaborating with NCEP.

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

30



**Table**

**Table 1: Crucial air quality indices for selected pollution cases in Beijing.**

| Selected pollution cases | Year/Month/Day | PM$_{2.5}$ (μg/m$^3$) | PM$_{10}$ (μg/m$^3$) | PM$_{2.5}$/ PM$_{10}$ | AQI |
|---|---|---|---|---|---|
| Explosive fine particle growth period | 2015/11/28 | 258 | 254 | 1.02 | 308 |
| | 2015/11/29 | 252 | NaN | NaN | 302 |
| Criterion: | 2015/11/30 | 343 | 550 | 0.62 | 450 |
| AQI>300; | 2015/12/01 | 464 | 454 | 1.02 | 476 |
| | 2015/12/22 | 297 | 427 | 0.70 | 347 |
| PM$_{2.5}$ > 250 μg/m3; | 2015/12/23 | 255 | 298 | 0.86 | 305 |
| PM$_{2.5}$/PM$_{10}$ > 0.70 (except on Nov. 30[th], 2015) | 2015/12/25 | 477 | 510 | 0.94 | 485 |
| | 2015/12/26 | 265 | 381 | 0.70 | 315 |
| | 2015/12/29 | 279 | 338 | 0.83 | 329 |
| | 2016/01/02 | 266 | 299 | 0.89 | 316 |
| Dust-dominated process | 2016/03/05 | 58 | 290 | 0.20 | 170 |
| | 2016/03/06 | 73 | 182 | 0.40 | 116 |
| Criterion: | 2016/03/28 | 70 | 195 | 0.36 | 123 |
| PM$_{10}$ > 150 μg/m3; | 2016/03/31 | 149 | 239 | 0.62 | 199 |
| | 2016/04/09 | 54 | 245 | 0.22 | 148 |
| PM$_{2.5}$/PM$_{10}$ ≤ 0.40 (except on Mar. 31[st], 2016) | 2016/04/10 | 41 | 192 | 0.21 | 121 |
| | 2016/05/05 | 62 | 153 | 0.40 | 102 |
| | 2016/05/06 | 57 | 182 | 0.31 | 116 |



**Figures**

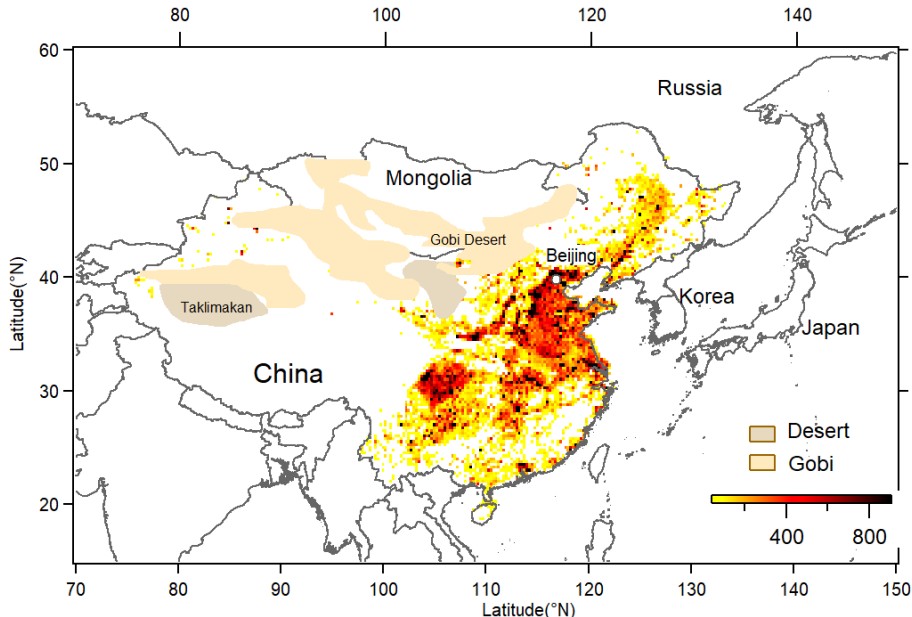

**Figure 1: Geographical location of the observation site in Beijing, PM$_{2.5}$ emissions in China and the location of major deserts and the Gobi in East Asia.**



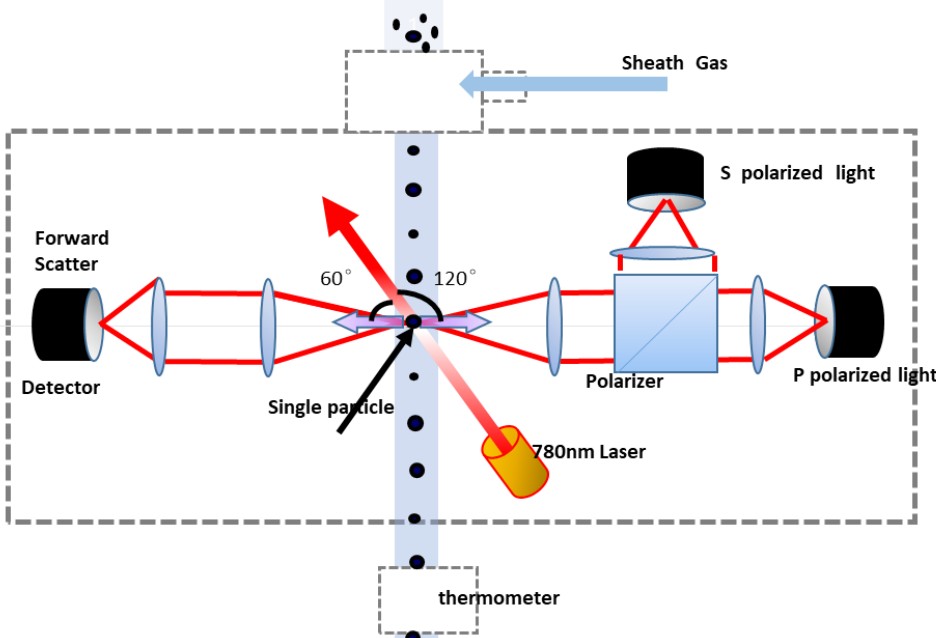

**Figure 2: Schematic diagram of the Polarization Optical Particle Counter.**



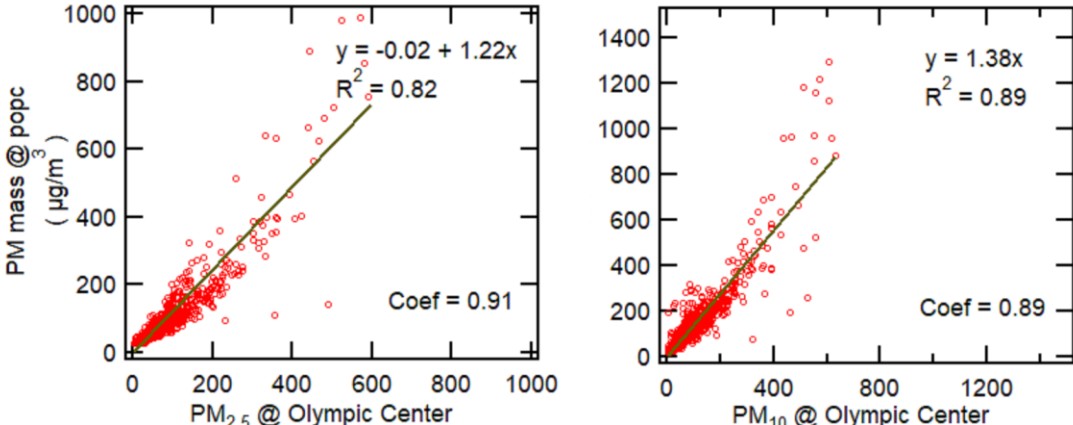

**Figure 3: Comparison of the POPC inverted particle concentration and real-time observation data from Olympic Center state control site.**







**Figure 4: Time series of (a) volume size distributions and relative humidity (gray solid line), (b) size-resolved depolarization ratio (Green solid line: DR = 0.1), and (c) hourly and monthly averaged depolarization ratios for fixed-size particles: Dp = 1 μm and 5 μm from November 29ᵗʰ, 2015 to July 29ᵗʰ, 2016. Error bars for monthly averaged DR in Figure 4c depict the monthly averaged deviation using the equation: V_adev $= \frac{1}{n}\sum_{i=0}^{n}|Yi - \overline{Y}|$ (n: number of points; $\overline{Y}$: average DR). The anthropogenic pollution case, polluted dust case and pure Asian dust case analyzed in this paper are denoted as "A", "B" and "C", respectively.**





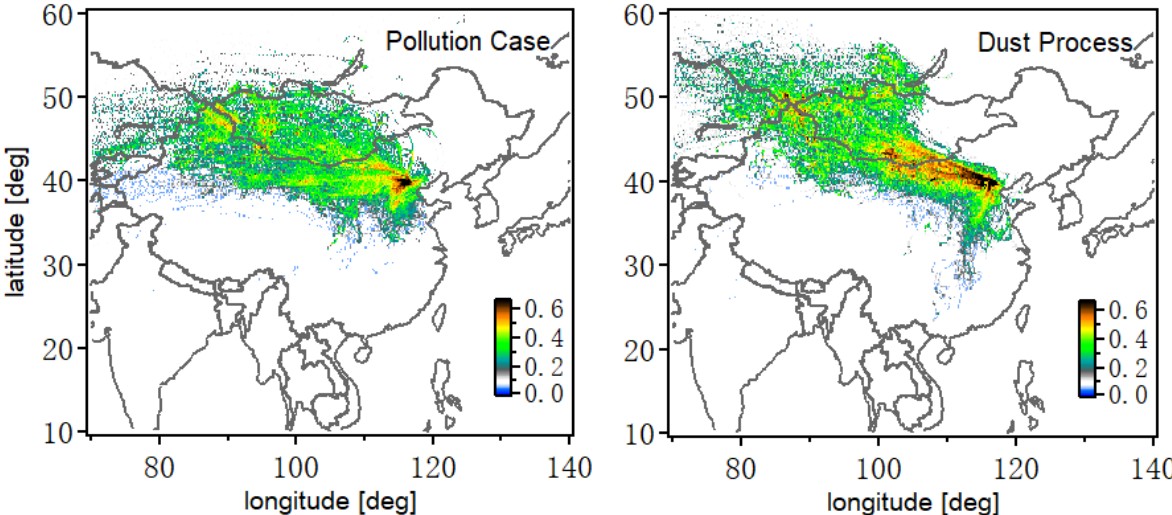

**Figure 5: Proportion of different directions from which the air mass over Beijing originated in varying pollution types: severe anthropogenic pollution period (Air Quality Index > 300 μg/m³ ; PM₂.₅ > 250 μg/m³ and PM₂.₅/PM₁₀ > 0.70) in the winter (left), and dust-dominant pollution case (PM₁₀ > 150 μg/m³; PM₂.₅/PM₁₀ < 0.40) in the spring (right). The red ellipse represents the major orientation from which the air mass came.**




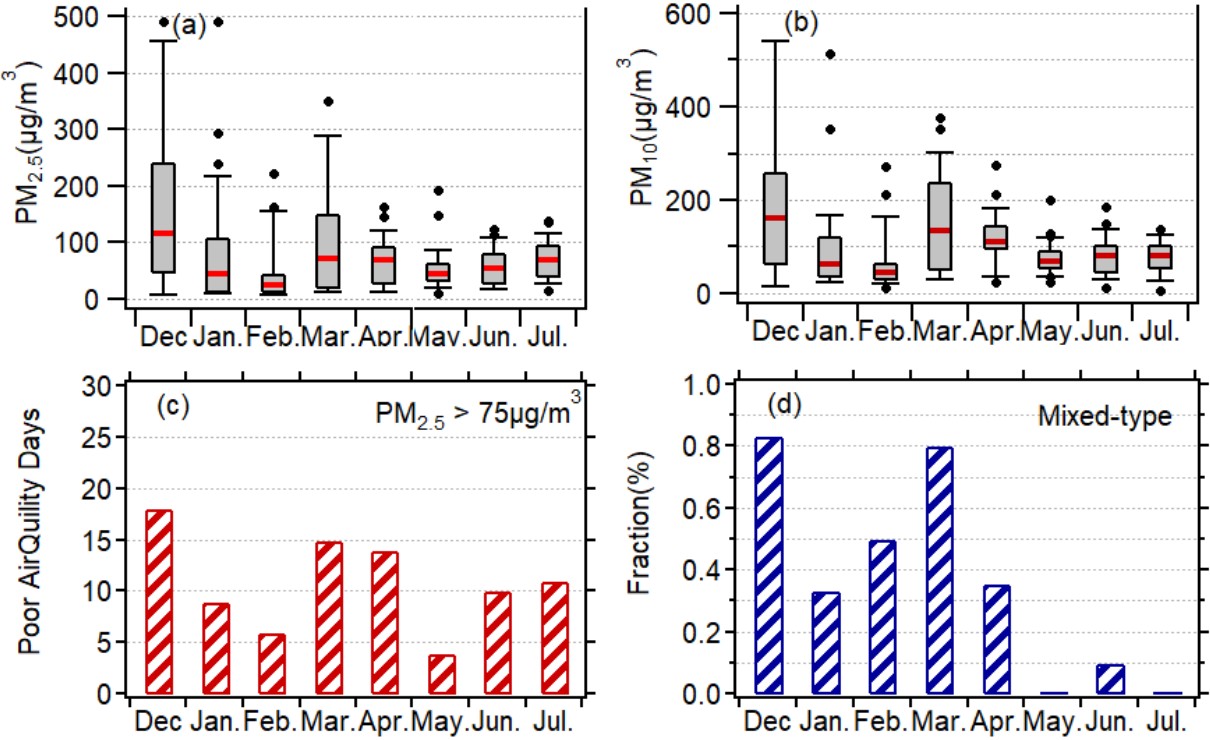

**Figure 6: Box and whisker plots of (a) mass concentration of PM$_{2.5}$, (b) PM$_{10}$ (the upper/lower ends of the bars bounds denote the 75$^{th}$/25$^{th}$ percentiles, the upper/lower whiskers denote the 90$^{th}$/10$^{th}$ percentiles and the lines in between signify the median values), (c) number of days with daily averaged mass concentrations of PM$_{2.5}$ exceeding 75 µg/m$^3$ (the secondary standard of Chinese Ambient Air Quality Standard of PM2.5), and (d) percentage of mixed-type pollution days in total poor air quality days over the observation period.**



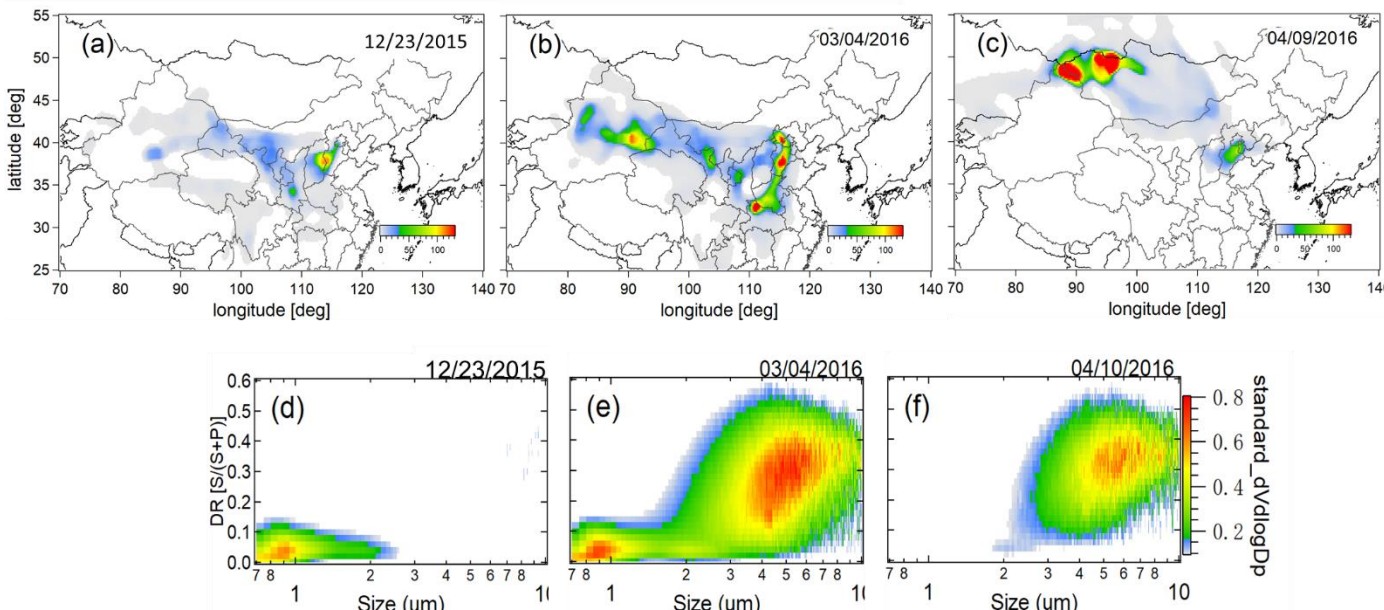

**Figure 7: Backward trajectories from Beijing calculated by the FLEXPART dispersion model for (a) the anthropogenic pollution case, (b) the mixed pollution period, (c) the pure mineral dust dominant case. Variation in the standard depolarization ratio as a function of particle size for corresponding episodes: (d), (e), and (f). Emission information for FLEXPART is explained in section 2.2, while information about the footprint and trajectory analysis is presented in the introduction.**





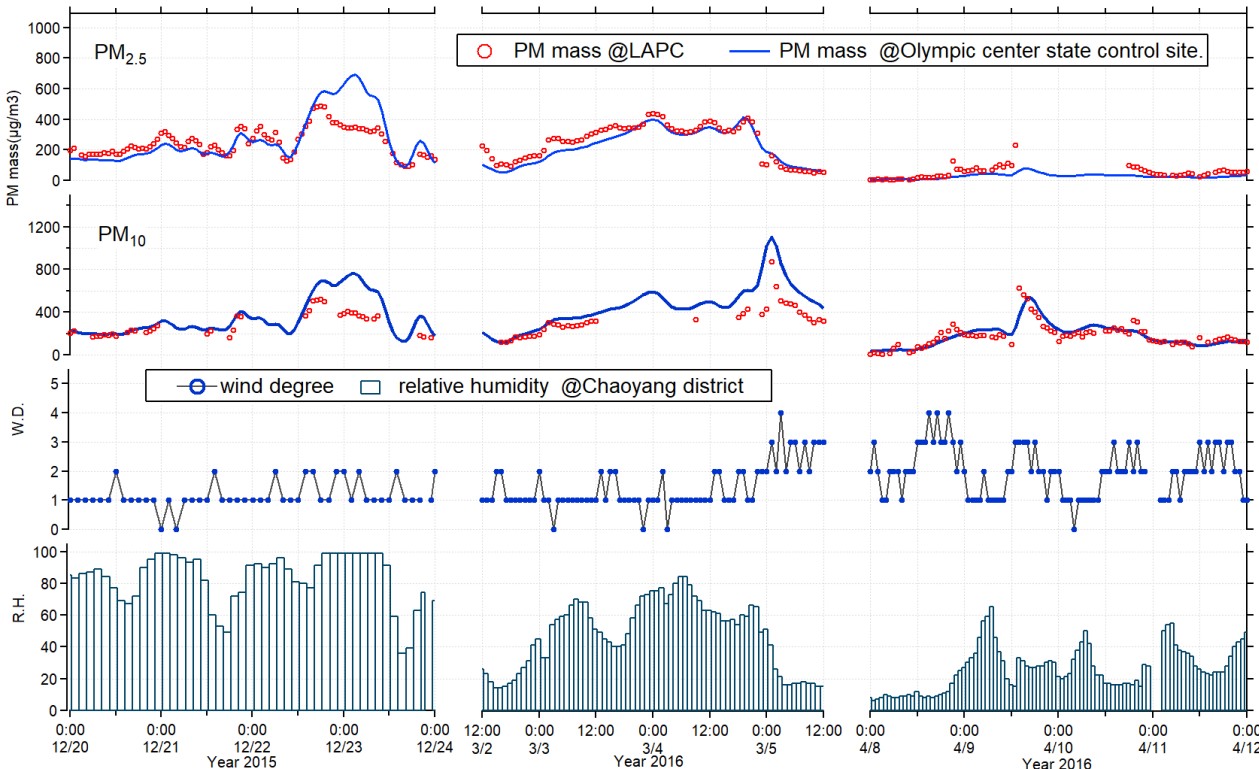

**Figure 8: Key meteorological factors and the air quality index during the mixing pollution period: PM$_{2.5}$; PM$_{10}$; W.D. represents the wind degree (corresponding table of specific wind degree and wind speed data is shown as Table S1 in the appendix); R.H. represents relative humidity (%).**




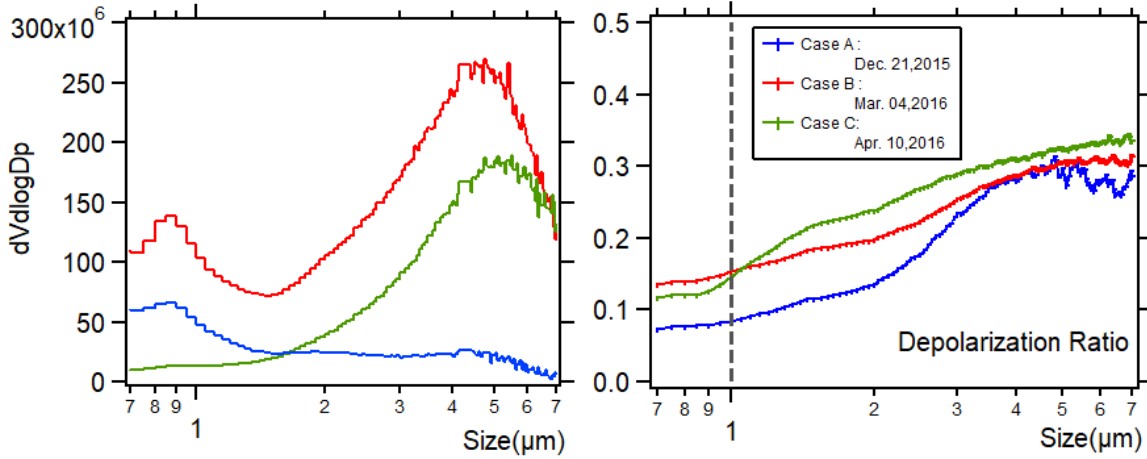

**Figure 9: Volume (left) and depolarization ratio (right) size distribution of aerosols observed in the study cases.**





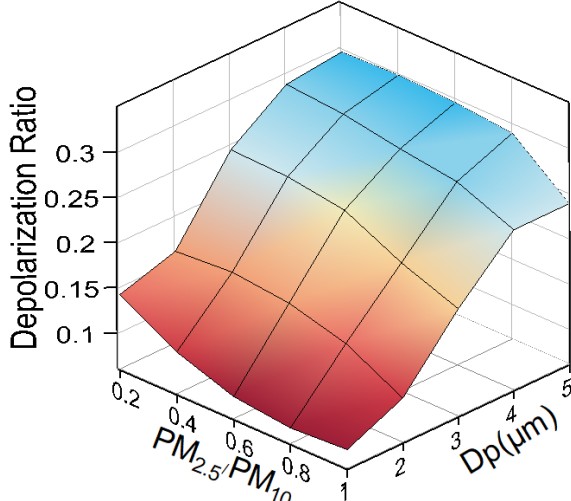

**Figure 10: Surface plot of the depolarization ratio as a function of particle size (from 1 µm to 5 µm) and PM₂.₅/PM₁₀ (from 0.2 to 1).**



**Figure 11. Scatter diagram of the relationship between the depolarization ratio of dust particles (at diameters of 3 µm and 5 µm), vaper content (RH) and PM$_{2.5}$/PM$_{10}$ in the air.**