# Peer review of "Variability of depolarization of aerosol particles in Beijing mega city: implication in interaction between anthropogenic pollutants and mineral dust particles"

_Atmospheric Chemistry and Physics, 2018_

## Referee Comment (RC1) · Anonymous Referee #1 · 26 Jul 2018

General Comments:

Tian et al. discuss aerosol particle polarization properties from November 29, 2015 to July 29, 2016 at Beijing. The dataset presented is interesting, particularly in showing particle polarization in anthropogenic pollution, mixed type pollution and pure Asian dust cases. Using an optical particle counter equipped with a polarization detection module is a new method to study mixing processes. However, the discussion in the current draft is insufficient for these data and results: a number of results/conclusions are not obtained from lab experiment or data prove, just derivate from published paper. I believe the article therefore should add data prove or lab experiment, and require major revisions.

Specific Comments:

1. Page 2, Line 2-3, only consumption of fossil fuel lead serious air pollution in East Asia? How about vehicle exhaust?

2. Page 2, Line 6, mineral dust particles only originated from natural sources? In the manuscript 3.4 part, the authors said emissions in famous mining areas in China can result in a huge amount of dust.

3. Page 3, Line 3, the authors should add more descriptions about depolarization of aerosol particles.

4. Page 3, Line 4-5, the authors said studies on the polarization characteristics are necessary, how about depolarization characteristics?

5. Page 3, Line 10-14, why key pollutants and meteorological data were obtained from two different sites? The key pollutants and meteorological data are the same in the two different sites?

6. Page 4, Line 28-29, why 150 m above ground level was chose? Why not is 100m or 200m? Why the simulation time of the backward trajectory is 5 days, not 3 days or 2 days?

7. Page 5, Line 9-11, What about the sources of particle size in 1 μm – 4 μm? A number of published papers said that $PM_{2.5}$ also mostly derived from anthropogenic sources. The reference of Kurosaki and Mikami, (2003) is a bit old.

8. Page 5, Line 24-26, the depolarization ratios of fine mode particles in summer are not lower than any other season? Why the authors emphasized depolarization ratios of particles with an optical diameter of 2 μm are lower than any other season?

9. Page 6, Line 4-7, wind speed, biomass burning or other factors have no impact on high depolarization ratio? Only transport lead high depolarization ratio?

10. Page 6, Line 7-10, have the authors ruled out other factors (such as wind speed, biomass burning)? These factors also can lead high $PM_{2.5}$.

11. Page 6, Line 30-31, the authors said "Mixed type pollution was also necessary for substandard days in Jan. (44%), 30 Feb. (50%), and Apr. (35%)". But in Page 5, Line 17-18, the authors said mixed type pollution is "B" (March 3rd to 4th, 2016). And please clearly describe how to decide mixed type pollution.

12. Page 7, Line 7-9, the authors obtained the conclusion is based on the report in published papers, not obtained from the authors' observation or lab experiment.

So possibly the conclusion didn't confirm.

13. Page 7, Line 25-26, why the authors confirm that increasing $PM_{2.5}$ level in case A was caused by weak diffusion conditions and high relative humidity? How about other aspects, such as emissions?

14. Page 7, Line 29-33, Fig, 7b didn't present height of air parcel, and in the paper, the authors also didn't present descriptions about height of air parcel in the draft. Why the authors can get conclusion that emissions in mining areas can lead a huge amount of dust? And in the paper, the descriptions about emissions in mining areas in March 3 to 4 are also not given.

15. Page 8, Line 6-9, please connected the cited part of Itahashi et al. (2010) with your study.

16. Page 8, Line 15, please give data prove or corresponding references.

17. Page 8, Line 16-17, how does the author judge no anthropogenic pollution emissions in case C, and obtain a pure dust process? In the paper, there are no data prove.

18. Page 9, Line 21-22, why the authors use $PM_{2.5}/PM_{10}$ ratio? In 3.1 part, the authors said the fine mode (<1 μm), the coarse mode (4 μm – 8 μm). Why the authors didn't use $PM_{1.0}$, $PM_{4.0-8.0}$?

19. Page 9, Line 25-26, generally we firstly presented some evidences, such as data prove or results in published paper, then derivated possible conclusion.

20. Page 9, Line 29-31, the result was obtained by the authors' lab experiment? If no, please add data prove or references.

21. Page 10, Line 3-20, please give corresponding data prove or lab experiment, then obtain conclusions.

Minor Comments:
1. Page 5, Line 11-13, "The growth of fine mode particles mainly stems from high anthropogenic emissions during heating seasons." confused with "The distinct growth period of fine mode particles was usually induced by an increase in relative humidity (i.e., RH)". Please afresh write the two sentences.

2. Page 6, Line 31, what is JJ? Please explain it.

3. Page 7, Line 3,4, NOx should be $NO_x$.

4. Page 8, Line 6, please clearly point out "dominant mechanisms". What were dominant mechanisms?

5. Page 8, Line 10-11, the authors said the first processes (22st -23rd, Dec. 2015) and the third process (9th, Apr. 2016). But in Page 5, Line 17-18, the authors said "A" (December 22nd to 23rd, 2015), "B" (March 3rd to 4th, 2016) and "C" (April 9th, 2016), respectively. Please unite.

6. Page 8, Line 15, please added data descriptions or references for "primary anthropogenic pollution emissions in this period were also lower compared to the winter time".

7. Page 9, Line 23-25, please added references.

8. Figure 7f, the date is 04/10/2016, not 04/09/2016?

9. Page 9, Line 24-25, please added references.

---

## Referee Comment (RC2) · Anonymous Referee #2 · 27 Aug 2018

The authors present a study of aerosol particles in Beijing, China, with a polarization optical particle counter (POPC). They present 8 months of statistics and 3 case studies of polluted conditions, dust conditions and a dust-pollution mixture. Especially the results presented in Fig. 7 are of interest, because they give insight in the mixing process of dust and pollution and help in the discussion about external and internal mixture.

While reading the manuscript, I was missing the main new points. In the revision process, I would strongly recommend to strengthen the impact of the study: the separation

of dust and pollution with a POPC and the discussion about internal or external mixture. Then, this can be applied to the statistics collected over an East Asian megacity, which lies in a global hot spot of dust and pollution mixtures.

The main point, why I have to reject the paper is that the retrieved depolarization values are questionable. I further checked the publication by Kobayashi et al., AE 2014, and I could not find any information on the calibration process of the depolarization ratio measured with the POPC. Especially the depolarization ratio can be influenced by multiple factors. I am sure, that the authors developed a calibration process to check the trustworthiness of the measured data, but it is missing in the publication. Also there are no comparisons of the retrieved depolarization ratios at 120° scattering angle to existing measurements or simulations of spherical or non-spherical particles. The uncertainties of the depolarization ratio values are not given. These points are essential to present trustworthy results. And as the depolarization ratio is one of the main quantities measured in this study, I have to insist on a proper calibration. Therefore, I recommend resubmitting the paper after clarifying the uncertainties of the depolarization ratio.

Further comments for improving the manuscript:

- Add an outline of the paper at the end of the introduction

- Mention, which size range is covered with your optical particle counter.

- The term "fine mode particles" normally refers to particles with a diameter smaller than 1 $\mu$m. PM1 data would be helpful to assess the fine mode particles.

- State clearer the difference between the depolarization ratio used in lidar studies (s-polarized to p-polarized ratio at 180° backscatter) and the depolarization ratio used in the POPC study (s-polarized to s+p polarized ratio at 120° backscatter). To add value to the lidar studies by using single particle analyses, it would be better to use the same depolarization ratio or at least to present a method to convert the POPC depolarization ratio to a lidar retrieved depolarization ratio. But I dare, that this would

be rather complicated for non-spherical particles.

- Add the distance and the direction (north, east, . . .) to the description of the two additional measurements sites (Olympic Sport Centre state control site and meteorological station) and not just the coordinates. It makes it easier for the reader.
* * *

---

## Author Comment (AC1) · 8 Oct 2018

**Reply to the comments of anonymous reviewer #1 on manuscript entitled "Variability of depolarization of aerosol particles in Beijing mega city: implication in interaction between anthropogenic pollutants and mineral dust particles"**

The authors appreciate very much for reviewing our manuscript and your insight comments. As suggested, we carefully revised the manuscript thoroughly according to the valuable advices, as well as proof-read the manuscript to minimize typographical, grammatical, and bibliographical errors.

**General Comments:**

*Tian et al. discuss aerosol particle polarization properties from November 29, 2015 to July 29, 2016 at Beijing. The dataset presented is interesting, particularly in showing particle polarization in anthropogenic pollution, mixed type pollution and pure Asian dust cases. Using an optical particle counter equipped with a polarization detection module is a new method to study mixing processes. However, the discussion in the current draft is insufficient for these data and results: a number of results/conclusions are not obtained from lab experiment or data prove, just derivate from published paper. I believe the article therefore should add data prove or lab experiment, and require major revisions.*

**Reply:** We fully agree with the reviewer's suggestions that simultaneous measurement of chemical compositions of particles and lab experiment evaluations will benefit the understanding of the aerosol's physical changes during mixed pollution. The change in aerosol's depolarization ratio can only represent that overall characteristics of the surface of dust aerosols, not the chemical reactions on it. As the reviewer suggested, we add more observational and experimental results in laboratory to clarify the chemical process happened during mix pollution, and the depolarization ratio of particles with meteorological factors, $PM_{2.5}$ and $PM_{10}$ concentrations, and aerosol chemical compositions are carefully discussed. To respond to the reviewer's major concerns, we made thoroughly revisions and corrections according to all the insight comments of the reviewers. Besides, more crucial/supporting information and analysis will be added in the revised manuscript, as follows:

(1) A detailed description and specification of calibration system of POPC and calibration results in this study will be added in the revised manuscript.

(2) Comparison between observation and the simulation result with T-matrix method on randomly-oriented Voronoi aggregation and elongated ellipsoid particles will be added.

(3) A comparison result of measurement of size-resolved number concentration of particles between POPC and KC52 optical particle counter will be added in the revised manuscript.

(4) Comparison of observation results between POPC measurement and LIDAR observation during several dust events in Beijing will be discussed. The cross validation on the dust events provide more direct evidence of variation in the depolarization properties of dust particles.

Besides, to make the conclusions more pertinent to this study, we revised the expression and concise the discussion about specific chemical composition and reaction. Our conclusions were useful for understanding the long-term depolarization property of aerosols in East Asia, interaction between anthropogenic pollutions and mineral dust in East Asia and their impacts on local air quality, and the sensitive relation between particle morphology and meteorological parameter. This long-term study in

Beijing motivate revisit on decades of Lidar data in dust-pollution interactions, and it indicates the necessity of a reliable optical model of internally mixed polluted dust for a detailed analysis of polarization remote sensing observations.

The initial purpose of this observational study is to investigate the long-term morphology properties of particles in Beijing because the North China region has long experienced seasonal haze and dust events due to the special geographical location and industrial development stage. As far as we know, due to limit of observation techniques, there was no studies reported on long-term variability of morphological characteristics of aerosols in north China till now. Alternatively, depolarization ratio of single particle is an applicable parameter indicating of general irregularity of aerosol particles. For this reason, an eight-month field measurement was performed at an urban site in Beijing megacity, we detected single particle depolarization ratio properties using a new developed beach-top instrument. Take advantage of meteorology/pollution data and trajectory analysis during the same period, the relationship between morphology change of particles and meteorological factors, especially water content in the atmosphere, was investigate. As pointed out, electro-microscopy inspection and composition results would provide direct evidence of chemo-physical changes of the particles, we will add more supporting information in the revised manuscript. The knotty point is that, single particle-based laboratory inspections need passive samples with variable time scales of 1 min to several days, it is difficult to have enough samplings to represent the long-term variability of aerosols statistically. We will follow reviewer's advice to conduct single-particle based electro-microscopic study in our next study. Chemical compositions of particles is helpful for understanding the possible chemical reaction during observation period. Unfortunately, we didn't have concurrent measurement during the experiment.

According to long-term real-time measurements of aerosol particle composition in Beijing in previous study, the seasonal variations of secondary inorganic aerosol (SIA, i.e., sulfate C nitrate C ammonium) concentrations were not significant, contributions of SIA were observed in summer (57 –61 %) and in winter (43 – 46 %) in fine mode aerosols (Sun et al., 2015). Although we did not have *in situ* measurement in this study in Beijing, a high $PM_{2.5}/PM_{10}$ ratio still indicates that the components of soluble inorganic salts was greatly possible to be relatively high, and in the case of high humidity. Therefore, morphology and corresponding depolarization ratio of particles should be affected. As expected, I did find a similar correlation between the ratio of $PM_{2.5}/PM_{10}$ and the depolarization ratio of coarse-mode aerosols, proving the great possibility that the inorganic salts might be trapped by large particles, or the occurrence of reaction between reactive gas and alkaline dust particles. At present, depolarization ratio is deemed as the most applicable method to rapidly or simultaneously distinguish types or mixing state of pollutions. Such characteristic are widely used by satellite-borne remote sensing and ground-based Lidar observations (Burton et al., 2012; Shimizu et al., 2004a; Shimizu et al., 2004b). In this study, combining analysis with LIDAR measurement, we hope to find an alternative and relatively simple indicator, such as the ratio of $PM_{2.5}/PM_{10}$, to estimate the possibility of morphology change of particles in the coarse-mode. In the manuscript, we revised the expression and weaken the discussion about specific chemical composition and reaction. In our next research, we would like to follow reviewer's suggestions to perform comprehensive measurements on the dependence of both morphology properties and chemical information of particles in different mixing state.

Sun, Y. L., Wang, Z. F., Du, W., Zhang, Q., Wang, Q. Q., Fu, P. Q., Pan, X. L., Li, J., Jayne, J., and Worsnop, D. R.: Long-term real-time measurements of aerosol particle composition in Beijing, China: seasonal variations, meteorological effects, and source analysis, Atmos Chem Phys, 15, 10149-10165,

10.5194/acp-15-10149-2015, 2015.

Burton, S. P., Ferrare, R. A., Hostetler, C. A., Hair, J. W., Rogers, R. R., Obland, M. D., Butler, C. F., Cook, A. L., Harper, D. B., and Froyd, K. D.: Aerosol classification using airborne High Spectral Resolution Lidar measurements – methodology and examples, Atmospheric Measurement Techniques, 5, 73-98, 10.5194/amt-5-73-2012, 2012.

Shimizu, A., Sugimoto, N., Matsui, I., Arao, K., Uno, I., Murayama, T., Kagawa, N., Aoki, K., Uchiyama, A., and Yamazaki, A.: Continuous observations of Asian dust and other aerosols by polarization lidars in China and Japan during ACE‐Asia, Journal of Geophysical Research: Atmospheres, 109, 2004a.

Shimizu, A., Sugimoto, N., Matsui, I., Arao, K., Uno, I., Murayama, T., Kagawa, N., Aoki, K., Uchiyama, A., and Yamazaki, A.: Continuous observations of Asian dust and other aerosols by polarization lidars in China and Japan during ACE-Asia, Journal of Geophysical Research-Atmospheres, 109, 10.1029/2002jd003253, 2004b.

**Specific Comments.**

1. *Page 2, Line 2-3, only consumption of fossil fuel lead serious air pollution in East Asia? How about vehicle exhaust?*

   **Reply:** We agree to the reviewer's comments. On-road vehicle exhaust plays an important role in formation of urban air pollution. We revised the expression in the manuscript as following, "Rapid economic growth in East Asia has caused serious air pollution in the past decades owing to the substantial consumption of fossil fuel and urbanization processes. Anthropogenic emissions (industrial, traffic-related, biomass burning etc.) lead to huge emits of pollutant gases and primary aerosols (Akimoto, 2003; Akimoto et al., 2006; Kurokawa et al., 2013; Li et al., 2017)

   Akimoto, H.: Global air quality and pollution, Science, 302, 1716-1719, 10.1126/science.1092666, 2003.

   Akimoto, H., Ohara, T., Kurokawa, J.-i., and Horii, N.: Verification of energy consumption in China during 1996–2003 by using satellite observational data, Atmospheric Environment, 40, 7663-7667, 10.1016/j.atmosenv.2006.07.052, 2006.

   Kurokawa, J., Ohara, T., Morikawa, T., Hanayama, S., Janssens-Maenhout, G., Fukui, T., Kawashima, K., and Akimoto, H.: Emissions of air pollutants and greenhouse gases over Asian regions during 2000–2008: Regional Emission inventory in ASia (REAS) version 2, Atmos Chem Phys, 13, 11019-11058, 10.5194/acp-13-11019-2013, 2013.

   Li, M., Zhang, Q., Kurokawa, J.-i., Woo, J.-H., He, K., Lu, Z., Ohara, T., Song, Y., Streets, D. G., Carmichael, G. R., Cheng, Y., Hong, C., Huo, H., Jiang, X., Kang, S., Liu, F., Su, H., and Zheng, B.: MIX: a mosaic Asian anthropogenic emission inventory under the international collaboration framework of the MICS-Asia and HTAP, Atmos Chem Phys, 17, 935-963, 10.5194/acp-17-935-2017, 2017.

2. *Page 2, Line 6, mineral dust particles only originated from natural sources? In the manuscript 3.4 part, the authors said emissions in famous mining areas in China can result in a huge amount of dust.*

**Reply:** The statement in section 3.4 made a misleading, we will revise the expression in the revised manuscript. What we want to express is that mining areas produce huge amount of pollutant gases $SO_2$, CO, as well as $NO_x$ related to vehicle exhaust in north China. In case B, there was a convergent air mass from natural dust sources (desert and Gobi) and mining areas in Beijing, according to FLEXPART trajectory analysis. It lead to a huge amount of dust mixed with pollutant-rich gaseous atmosphere in the Beijing area. As the reviewer pointed out, mineral dust particles in city environment might also be emitted from on-road vehicle and construction dust. These fugitive dust usually have a large diameter and a short lifetime due to rapid gravity settlement velocity. In this paper, we focus on typical, severe, synoptic dust events, which was usually nature sources. The sentence has been revised as "Mineral dust particles in the atmosphere also have a detrimental impact on air quality by reducing visibility and bolstering concentrations of inhalable particulate matter (Goudie, 2014)"

Goudie, A. S.: Desert dust and human health disorders, Environ Int, 63, 101-113, 10.1016/j.envint.2013.10.011, 2014.

3.  *Page 3, Line 3, the authors should add more descriptions about depolarization of aerosol particles.*

**Reply:** We will add more descriptions about depolarization of particles. Volume depolarization ratio of particle groups is widely used in LIDAR detection to distinguish the types of aerosol particles. The volume depolarization information of target air parcel just represents an overall depolarization. In this study, we use a new instrument to measure real-time variations in single particle depolarization ratio. This measurement can avoid the influences from the presence of substantial amount of small spherical particles. As mentioned, direction of polarization of scattering light was identical with the incident light for spherical particle; however, for the non-spherical particles, the direction of polarization of light will be deviated. According to such phenomenon, mineral dust and anthropogenic pollution dominant particles can be distinguished. Sugimoto et al. (2015) found that backscattering depolarization ratio in polluted dust was smaller compared to pure Asia dust, Pan et al. (2017) reported the vital role in observed depolarization ratio decrease that calcium nitrate played via coating process and hygroscopic growth on the surface of dust particles. In a word, single particle-based depolarization ratio is applicable for quantitatively investigating the evolution of the mixing of dust particles during their transport. As suggested by the reviewer, we will add and clarify the relevant explanation about depolarization ratio in the revised manuscript.

Sugimoto, N., Nishizawa, T., Shimizu, A., Matsui, I., and Kobayashi, H.: Detection of internally mixed Asian dust with air pollution aerosols using a polarization optical particle counter and a polarization-sensitive two-wavelength lidar, Journal of Quantitative Spectroscopy & Radiative Transfer, 150, 107-113, 10.1016/j.jqsrt.2014.08.003, 2015.

Pan, X., Uno, I., Wang, Z., Nishizawa, T., Sugimoto, N., Yamamoto, S., Kobayashi, H., Sun, Y., Fu, P., Tang, X., and Wang, Z.: Real-time observational evidence of changing Asian dust morphology with the mixing of heavy anthropogenic pollution, Sci Rep, 7, 335, 10.1038/s41598-017-00444-w, 2017.

4.  *Page 3, Line 4-5, the authors said studies on the polarization characteristics are necessary, how about depolarization characteristics?*

**Reply:** To avoid misleading, we will use depolarization ratio of particle in the whole context. Actually, identical laser hit particles and producing different polarized light signals. The different is caused by particle morphology and best described by the ratio of "depolarization". I will make corresponding corrections in revised manuscript.

5. *Page 3, Line 10-14, why key pollutants and meteorological data were obtained from two different sites? The key pollutants and meteorological data are the same in the two different sites?*

**Reply:** We select the nearest state-controlled environmental monitoring station and weather station to the observation experiment site to support this study. As known, the horizontal scale of an air mass is 10 kilometers to hundreds of kilometers, the synoptic condition of the air mass (pollutant concentration, humidity and temperature) in city scale are normally the similar (as shown in Figure 1), though differences existed due to possible local sources. To better describe the two additional measurements sites (Olympic Sport Centre state control site and meteorological station) and makes it easier for the readers, we change the expression as: "Besides, air quality of community in the observation period were reviewed according to observed key pollutants, including the mass concentrations of $PM_{2.5}$ and $PM_{10}$ came from the Olympic Sport Centre state control station (Longitude: 116.40E; Latitude: 39.98N; 2.7 km northeast of LAPC). During this period, three pollution events were chosen to investigate the interactions between dust particles and pollutants, together with an analysis of meteorology parameters obtained from the Beijing Chaoyang district meteorological station (Longitude: 116.48E; Latitude: 39.95N; 9.5 km southeast of LAPC)."

[Figure]

Figure 1. Time serious of particle mass concentration of PM$_{2.5}$ and PM$_{10}$, and meteorology factors (relative humidity and temperature) in different site in Beijing area.

6.  *Page 4, Line 28-29, why 150 m above ground level was chose? Why not is 100m or 200m? Why the simulation time of the backward trajectory is 5 days, not 3 days or 2 days?*

**Reply:** We performed backward trajectory simulation starting from height of 150 m, because our observation is on a top of two-story building (~10 meter high above the ground), and urban canopy of Beijing city was estimated to be less than 200 m. Considering that we use trajectories to indicate synoptic transport of dust aerosols, a starting height of 150 m is chosen. As a matter of fact, the trajectories starting from 150 m and 200 m are similar, and did not change our major verdict. A weather process is generally last for 3-7 days. We choose 5 days simulation because it can better reflect the long-distance transmission of particles. 2 or 3 days simulation may only reflect the local impact, but the typically continuous dust process has a great probability to be influenced by long-distance transmission in North China.

7. *Page 5, Line 9-11, What about the sources of particle size in 1 μm – 4 μm? A number of published papers said that PM2.5 also mostly derived from anthropogenic sources. The reference of Kurosaki and Mikami, (2003) is a bit old.*

**Reply:** The particle with aerodynamic diameter less than 2.5 μm is defined as fine mode, and the particle size between 2.5 – 10 μm is defined as coarse mode. As mentioned, the particles between 1 – 4 μm consisted of both anthropogenic pollutants such as sulfate, nitrate and organic matters and mineral dust particle from both nature and anthropogenic sources. Numbers of previous studies indicate that the mode diameter of dust particle in North China is normally about 5 μm. It was consistent with our measurement of size distribution which showed a broad peak between 4 – 8 μm. We will revised our manuscript "… while the particle in coarse mode can be attributed to mineral dust particles from both nature sources, in particular in spring season (Li et al., 2012)"

Li, J., Wang, Z., Zhuang, G., Luo, G., Sun, Y., and Wang, Q.: Mixing of Asian mineral dust with anthropogenic pollutants over East Asia: a model case study of a super-duststorm in March 2010, Atmos Chem Phys, 12, 7591-7607, 10.5194/acp-12-7591-2012, 2012.

8. *Page 5, Line 24-26, the depolarization ratios of fine mode particles in summer are not lower than any other season? Why the authors emphasized depolarization ratios of particles with an optical diameter of 2 μm are lower than any other season?*

**Reply:** As reminded, the depolarization ratios of all fine mode particles in summer were lower than other seasons, as shown in Figure 4 in the manuscript. In the present study, we chose the particle with optical diameter of 2 μm to indicate the decreasing trend. In summer time, Beijing was effected by East Asian monsoon and prevails southeast winds which bring more high humid air and precipitation. High relative humidity could lead to deliquesce and hygroscopic growth happened on soluble component in aerosols. Depolarization ratio of 2 μm particles was separately mentioned as an example. We will clarify this point in the revised manuscript.

9. *Page 6, Line 4-7, wind speed, biomass burning or other factors have no impact on high depolarization ratio? Only transport lead high depolarization ratio?*

**Reply:** We consent to the reviewer's comment that depolarization properties of different aerosol types are different. For instance, the soot particles from biomass burning normally have complex chain-like morphology which was distinct from spherical particles. In this study, the depolarization ratio actually reflect the type of particle: mineral dust, and anthropogenic pollutant or sea salt aerosols (Kobayashi et al., 2014). We will revise the interpretation in the manuscript as "For the days

when coarse mode aerosols dominated which happened mostly in spring-time, the air mass mainly originated from northwest areas, transnational Inner Mongolia and Gobi Desert that are all natural sources of dusts. This trajectory pattern represent comparatively larger dust loading in the atmosphere, partially explains why the monthly averaged depolarization ratio of all particle size in Beijing can reach the highest 0.26 in March, a mostly windy and dry month with comparatively frequent dust events (Shao and Dong, 2006)."

Kobayashi, H., Hayashi, M., Shiraishi, K., Nakura, Y., Enomoto, T., Miura, K., Takahashi, H., Igarashi, Y., Naoe, H., Kaneyasu, N., Nishizawa, T., and Sugimoto, N.: Development of a polarization optical particle counter capable of aerosol type classification, Atmospheric Environment, 97, 486-492, 10.1016/j.atmosenv.2014.05.006, 2014.

Shao, Y., and Dong, C. H.: A review on East Asian dust storm climate, modelling and monitoring, Global and Planetary Change, 52, 1-22, 10.1016/j.gloplacha.2006.02.011, 2006.

10. *Page 6, Line 7-10, have the authors ruled out other factors (such as wind speed, biomass burning)? These factors also can lead high PM2.5.*

**Reply:** Yes, as pointed out by the reviewer, factors including wind speed and biomass burning also can lead high $PM_{2.5}$. Previous studies of Wang et al. (2013) and Chang and Zhan (2017) have demonstrated the necessary effect of weak wind speed on haze formation. Chen et al. (2017) and Cheng et al. (2014) have also demonstrated the importance of biomass burning high $PM_{2.5}$ levels by observing facts. It reflected that the biomass combustion and specific synoptic situation was of great importance in variation of $PM_{2.5}$. In this study, we forces on the depolarization properties in ambient air in Beijing area, and the trajectory analysis was to explain the seasonal change of depolarization ratio. Meteorology factors including wind speed, RH was discussed in 3.4 and 3.5 part. In our next study, we would like to quantify the relationship between biomass burning and $PM_{2.5}$ levels, and investigate the change and effect on depolarization ratio of ambient air.

Wang, Y., Yao, L., Wang, L., Liu, Z., Ji, D., Tang, G., Zhang, J., Sun, Y., Hu, B., and Xin, J.: Mechanism for the formation of the January 2013 heavy haze pollution episode over central and eastern China, Science China Earth Sciences, 57, 14-25, 10.1007/s11430-013-4773-4, 2013.

Chang, W., and Zhan, J.: The association of weather patterns with haze episodes: Recognition by PM2.5 oriented circulation classification applied in Xiamen, Southeastern China, Atmospheric Research, 197, 425-436, 10.1016/j.atmosres.2017.07.024, 2017b.

Chen, J., Li, C., Ristovski, Z., Milic, A., Gu, Y., Islam, M. S., Wang, S., Hao, J., Zhang, H., He, C., Guo, H., Fu, H., Miljevic, B., Morawska, L., Thai, P., Lam, Y. F., Pereira, G., Ding, A., Huang, X., and Dumka, U. C.: A review of biomass burning: Emissions and impacts on air quality, health and climate in China, Sci Total Environ, 579, 1000-1034, 10.1016/j.scitotenv.2016.11.025, 2017.

Cheng, Z., Wang, S., Fu, X., Watson, J. G., Jiang, J., Fu, Q., Chen, C., Xu, B., Yu, J., Chow, J. C., and Hao, J.: Impact of biomass burning on haze pollution in the Yangtze River delta, China: a case study in summer 2011, Atmos Chem Phys, 14, 4573-4585, 10.5194/acp-14-4573-2014, 2014.

11. *Page 6, Line 30-31, the authors said "Mixed type pollution was also necessary for substandard days in Jan. (44%), Feb. (50%), and Apr. (35%)". But in Page 5, Line 17-18, the authors said mixed type*

*pollution is"B" (March 3rd to 4th, 2016). And please clearly describe how to decide mixed type pollution.*

**Reply:** In this study, mixed type pollution means coexistence of anthropogenic pollutants and dust particles in the atmosphere. The mixed pollution days was calculated by daily $PM_{2.5} > 75$ μg/m$^3$ and $0.4 < PM_{2.5}/PM_{10} < 0.7$, for higher ratio of $PM_{2.5}/PM_{10}$ was generally ascribed to high contributions from combustion sources and secondary particles; lower ratios generally indicate significant contributions from primary sources such as re-suspended soil/road/construction dust, and mechanical activities (Xu et al., 2017). There was an error in the calculation result and we have made some modifications in the manuscript. We accept suggestions from the reviewer and will further explain the definition of mixed pollution in the revised manuscript for better understanding of readers.

Xu, G., Jiao, L., Zhang, B., Zhao, S., Yuan, M., Gu, Y., Liu, J., and Tang, X.: Spatial and Temporal Variability of the PM2.5/PM10 Ratio in Wuhan, Central China, Aerosol and Air Quality Research, 17, 741-751, 10.4209/aaqr.2016.09.0406, 2017.

12. *Page 7, Line 7-9, the authors obtained the conclusion is based on the report in published papers, not obtained from the authors' observation or lab experiment. So possibly the conclusion didn't confirm.*

**Reply:** We will revise our expression in revised manuscript as "On these grounds, we suspect that polluted dust induced heterogeneous photochemical reactions is a major factor in reducing Beijing's air quality, especially in spring-time, confirming the observational data."

13. *Page 7, Line 25-26, why the authors confirm that increasing PM2.5 level in case A was caused by weak diffusion conditions and high relative humidity? How about other aspects, such as emissions?*

**Reply:** The heating season in north China is from mid-November to mid-March of the following year. The pollutant emission in Beijing area in this period remained at a high level compared with any other seasons (Wang et al., 2015). Haze pollution can essentially be attributed to two aspects: pollutant emissions into the lower atmosphere and favorable meteorological conditions. In particular, meteorological conditions control the occurrence of haze pollution (Zhang et al., 2016; Wang et al., 2013; Jacob and Winner, 2009; Ilten and Selici, 2008; Chang and Zhan, 2017). High emission of primary pollutant precursors in winter time combine with stable weather conditions (weak diffusion conditions) and favorable conditions for liquid phase reaction of pollutant (high humid), resulting in the formation of secondary pollutants $PM_{2.5}$. And the clarity of the pollution process is usually accompanied by a strong cold air which from Siberia spills south into East Asia. We will make the reasons for high $PM_{2.5}$ clearer in the manuscript.

Wang, Q., Sun, Y., Jiang, Q., Du, W., Sun, C., Fu, P., and Wang, Z.: Chemical composition of aerosol particles and light extinction apportionment before and during the heating season in Beijing, China, Journal of Geophysical Research: Atmospheres, 120, 12708-12722, 10.1002/2015jd023871, 2015.

Zhang, Z., Zhang, X., Gong, D., Kim, S. J., Mao, R., and Zhao, X.: Possible influence of atmospheric circulations on winter haze pollution in the Beijing–Tianjin–Hebei region, northern China, Atmos Chem Phys, 16, 561-571, 10.5194/acp-16-561-2016, 2016.

Wang, Y., Yao, L., Wang, L., Liu, Z., Ji, D., Tang, G., Zhang, J., Sun, Y., Hu, B., and Xin, J.: Mechanism for the formation of the January 2013 heavy haze pollution episode over central and eastern China, Science China Earth Sciences, 57, 14-25, 10.1007/s11430-013-4773-4, 2013.

Jacob, D. J., and Winner, D. A.: Effect of climate change on air quality, Atmospheric Environment, 43, 51-63, 10.1016/j.atmosenv.2008.09.051, 2009.

Ilten, N., and Selici, A. T.: Investigating the impacts of some meteorological parameters on air pollution in Balikesir, Turkey, Environ Monit Assess, 140, 267-277, 10.1007/s10661-007-9865-1, 2008.

Chang, W., and Zhan, J.: The association of weather patterns with haze episodes: Recognition by PM2.5 oriented circulation classification applied in Xiamen, Southeastern China, Atmospheric Research, 197, 425-436, 10.1016/j.atmosres.2017.07.024, 2017.

14. *Page 7, Line 29-33, Fig, 7b didn't present height of air parcel, and in the paper, the authors also didn't present descriptions about height of air parcel in the draft. Why the authors can get conclusion that emissions in mining areas can lead a huge amount of dust? And in the paper, the descriptions about emissions in mining areas in March 3 to 4 are also not given.*

**Reply:** As suggested, we will add detailed simulation information in section 2.2 and section 3.4. We will revise the analysis for the case B, "… In case B, the air mass was a convergence of deviating northwest and south streams. The eastward-conveying air mass originated from the Taklimakan Desert in Sinkiang and channeled the Tengger Desert in Inner Mongolia, which were major source of dust in east Asia (Fu et al., 2012; Chen et al., 2013; Chen et al., 2017; Zhang et al., 2018); the south branch of the air mass past central Henan Province and southern Hebei Province where emission of gaseous pollutants from industries, residential and transport sources were significant (Huang et al., 2012;Zhao et al., 2012;Wang et al., 2014). Reactive gaseous pollutant mixed with mineral dust from west natural dust sources, result in a mixed pollution types (Figure 7e)."

Fu, P., Zhong, S., Huang, J., and Song, G.: An observational study of aerosol and turbulence properties during dust storms in northwest China, Journal of Geophysical Research: Atmospheres, 117, n/a-n/a, 10.1029/2011jd016696, 2012.

Chen, S., Huang, J., Zhao, C., Qian, Y., Leung, L. R., and Yang, B.: Modeling the transport and radiative forcing of Taklimakan dust over the Tibetan Plateau: A case study in the summer of 2006, Journal of Geophysical Research: Atmospheres, 118, 797-812, 10.1002/jgrd.50122, 2013.

Chen, S., Huang, J., Kang, L., Wang, H., Ma, X., He, Y., Yuan, T., Yang, B., Huang, Z., and Zhang, G.: Emission, transport, and radiative effects of mineral dust from the Taklimakan and Gobi deserts: comparison of measurements and model results, Atmos Chem Phys, 17, 2401-2421, 10.5194/acp-17-2401-2017, 2017.

Zhang, X.-X., Sharratt, B., Liu, L.-Y., Wang, Z.-F., Pan, X.-L., Lei, J.-Q., Wu, S.-X., Huang, S.-Y., Guo, Y.-H., Li, J., Tang, X., Yang, T., Tian, Y., Chen, X.-S., Hao, J.-Q., Zheng, H.-T., Yang, Y.-Y., and Lyu, Y.-L.: East Asian dust storm in May 2017: observations, modelling and its influence on Asia-Pacific region, Atmospheric Chemistry and Physics Discussions, 1-31, 10.5194/acp-2018-205, 2018.

Huang, X., Song, Y., Li, M., Li, J., Huo, Q., Cai, X., Zhu, T., Hu, M., and Zhang, H.: A high-resolution ammonia emission inventory in China, Global Biogeochemical Cycles, 26, n/a-n/a, 10.1029/2011gb004161, 2012.

Zhao, B., Wang, P., Ma, J. Z., Zhu, S., Pozzer, A., and Li, W.: A high-resolution emission inventory of primary pollutants for the Huabei region, China, Atmos Chem Phys, 12, 481-501, 10.5194/acp-12-481-2012, 2012.

Wang, L. T., Wei, Z., Yang, J., Zhang, Y., Zhang, F. F., Su, J., Meng, C. C., and Zhang, Q.: The 2013 severe haze over southern Hebei, China: model evaluation, source apportionment, and policy implications, Atmos Chem Phys, 14, 3151-3173, 10.5194/acp-14-3151-2014, 2014.

15. *Page 8, Line 6-9, please connected the cited part of Itahashi et al. (2010) with your study.*

**Reply:** We will make analysis more clearly with supporting information from the literature. "On 1200UTC 4 Mar 2016, the Beijing area was at the bottom of a transit low-pressure system. The temperature field lag behind the height field at 850 hpa, dense isotherm and strong wind shear represent cold air invasion at low-altitude over the East China area, this is also where the cold front wind zone was located (Figure S3b). At height 500 hpa, the geopotential height field showed a zonal circulation dominated pattern with temperature field be in same phase, indicating the cold air input mechanism at low altitude will be maintained (Figure S3a). On the map of sea-level pressure and observed meteorological elements at 1200UTC 4 March, precipitation and fog day and positive 3-hour surface pressure change caused by passed cold frond covered Eastern China and the Korean peninsula (Figure S3c). Lidar observation also captured the mineral dust impacting case, dust was clearly visible beginning on 00 am. March 4, to at an altitude of 3 - 4 km (Figure 9 a-b). The dust plume was then compressed by external force or gravitational sedimentation and decreased to the altitude < 1 km until morning of March 6. Itahashi et al. (2010) studied six cases of large-scale dust transport events occurring in May 2007 in Japan, found that four events corresponded to a "behind cold front" pattern. This pattern was characterized by the observation site locate at the bottom of a single low-pressure system and behind a well-developed cold front, atmospheric stratification like this will lead to both dust and pollutants trapped and mixed within the PBL. Suggest that there was great possibly that mineral dust mixed with reactive pollutant because of strong stratification's trapping effect in Case B.".

Itahashi, S., Yumimoto, K., Uno, I., Eguchi, K., Takemura, T., Hara, Y., Shimizu, A., Sugimoto, N., and Liu, Z. Y.: Structure of dust and air pollutant outflow over East Asia in the spring, Geophysical Research Letters, 37, n/a-n/a, 10.1029/2010gl044776, 2010.

16. *Page 8, Line 15, please give data prove or corresponding references.*

**Reply:** We would like to cite Wang et al., (2015) in the manuscript, and revise the sentence as "Considering the heating season was now over for a month, $PM_{2.5}$ that associated with high burning of fossil fuels for heating purpose was relatively weak (Wang et al., 2015)"

Wang, Q., Sun, Y., Jiang, Q., Du, W., Sun, C., Fu, P., and Wang, Z.: Chemical composition of aerosol particles and light extinction apportionment before and during the heating season in Beijing, China, Journal of Geophysical Research: Atmospheres, 120, 12708-12722, 10.1002/2015jd023871, 2015.

17. *Page 8, Line 16-17, how does the author judge no anthropogenic pollution emissions in case C, and obtain a pure dust process? In the paper, there are no data prove.*

**Reply:** We agree with the reviewer that we cannot describe case C as completely free of anthropogenic pollution due to lacking of chemical composition information. During the period, the daily $PM_{10}$ was 245 $\mu g/m^3$ and $PM_{2.5}$ account for 22% in $PM_{10}$ in April 9, and daily $PM_{10}$ was 192 $\mu g/m^3$ with $PM_{2.5}$ account for 21% in April 10. POPC analysis (Figure 7f in the manuscript) showed that particles in this process was concentrated in the particle size range larger than 3 $\mu m$, in which that contribution secondary formed aerosol particles was less important. During case C, air mass originated from gobi desert in Mongolia and transport rapidly by a strong wind. Recent filter-based observation (Pan et al., 2017) showed that the water-soluble compounds in coarse mode was insignificant, compared to crustal matters, during dust event. To avoid misleading, we will defined Case C as a typical dust-dominant period. We will make a more comprehensive interpretation of this conclusion in the revised manuscript according to the comments of reviewers.

Pan, X., Uno, I., Wang, Z., Nishizawa, T., Sugimoto, N., Yamamoto, S., Kobayashi, H., Sun, Y., Fu, P., Tang, X., and Wang, Z.: Real-time observational evidence of changing Asian dust morphology with the mixing of heavy anthropogenic pollution, Sci Rep, 7, 335, 10.1038/s41598-017-00444-w, 2017.

18. *Page 9, Line 21-22, why the authors use $PM_{2.5}/PM_{10}$ ratio? In 3.1 part, the authors said the fine mode (<1 $\mu m$), the coarse mode (4 $\mu m$ – 8 $\mu m$). Why the authors didn't use $PM_{1.0}$, $PM_{4.0-8.0}$?*

**Reply:** Sorry for the misleading. The particles with aerodynamic diameter less than 2.5 $\mu m$ is defined as fine mode, and the particles with diameter between 2.5 and 10 $\mu m$ is defined as coarse mode. The widely adopted commercial online instruments normally measure the mass concentrations of $PM_{2.5}$ and $PM_{10}$. For POPC measurement, the mass concentrations of particles of $PM_{1.0}$ and $PM_{4.0-8.0}$ could be reasonably calculated on the basis of size distribution of particles. However, this measurement at the state-control observatories are very limited in China, which make it difficult to estimate the variability of aerosol types. Alternatively, we adopted the ratio $PM_{2.5}/PM_{10}$ to identify the sources of primary pollutant in previous studies. A higher ratio was generally ascribed to anthropogenic related secondary particles (sulfate, nitrate etc.), and a lower ratio indicate significant contributions mainly re-suspended or fugitive mineral dust particles due to some mechanical processes (Chan et al., 2005; Pérez et al., 2008; Akyuz and Cabuk, 2009; Chan and Yao, 2008; Xu et al., 2017). In a typical dust period, the higher the ratio was, the more possibility that anthropogenic pollution and dust particles were mixing, and consequentially morphological changes. We agree with the reviewer's concerns and will add the explanations for using the $PM_{2.5}/PM_{10}$ ratio in the manuscript.

Chan, C. Y., Xu, X. D., Li, Y. S., Wong, K. H., Ding, G. A., Chan, L. Y., and Cheng, X. H.: Characteristics of vertical profiles and sources of PM2.5, PM10 and carbonaceous species in Beijing, Atmospheric Environment, 39, 5113-5124, 10.1016/j.atmosenv.2005.05.009, 2005.

Pérez, N., Pey, J., Querol, X., Alastuey, A., López, J. M., and Viana, M.: Partitioning of major and trace components in PM10–PM2.5–PM1 at an urban site in Southern Europe, Atmospheric Environment, 42, 1677-1691, 10.1016/j.atmosenv.2007.11.034, 2008.

Akyuz, M., and Cabuk, H.: Meteorological variations of PM2.5/PM10 concentrations and particle-associated polycyclic aromatic hydrocarbons in the atmospheric environment of Zonguldak, Turkey, J Hazard Mater, 170, 13-21, 10.1016/j.jhazmat.2009.05.029, 2009.

Chan, C. K., and Yao, X.: Air pollution in mega cities in China, Atmospheric Environment, 42, 1-42, 10.1016/j.atmosenv.2007.09.003, 2008.

Xu, G., Jiao, L., Zhang, B., Zhao, S., Yuan, M., Gu, Y., Liu, J., and Tang, X.: Spatial and Temporal Variability of the PM2.5/PM10 Ratio in Wuhan, Central China, Aerosol and Air Quality Research, 17, 741-751, 10.4209/aaqr.2016.09.0406, 2017.

19. *Page 9, Line 25-26, generally we firstly presented some evidences, such as data prove or results in published paper, then derivate possible conclusion.*

**Reply:** We will cite more relevant reference to support our conclusion in the revised manuscript as "Another possible reason was that the polluted air contains abundant $HNO_3$ (Liang et al., 2007; Shi et al., 2014; Bytnerowicz et al., 2016), deeper interaction between mineral dust and reactive acidic gases and the trapping process of atmospheric secondary inorganic salt modified the hydrophilic state of dust aerosols (Kelly and Wexler, 2005;Sullivan et al., 2009), lead to accelerated decrease of depolarization ratio at a relatively higher RH condition."

Liang, Q., Jaeglé, L., Hudman, R. C., Turquety, S., Jacob, D. J., Avery, M. A., Browell, E. V., Sachse, G. W., Blake, D. R., Brune, W., Ren, X., Cohen, R. C., Dibb, J. E., Fried, A., Fuelberg, H., Porter, M., Heikes, B. G., Huey, G., Singh, H. B., and Wennberg, P. O.: Summertime influence of Asian pollution in the free troposphere over North America, Journal of Geophysical Research, 112, 10.1029/2006jd007919, 2007.

Shi, Y., Chen, J., Hu, D., Wang, L., Yang, X., and Wang, X.: Airborne submicron particulate (PM1) pollution in Shanghai, China: chemical variability, formation/dissociation of associated semi-volatile components and the impacts on visibility, Sci Total Environ, 473-474, 199-206, 10.1016/j.scitotenv.2013.12.024, 2014.

Bytnerowicz, A., Hsu, Y. M., Percy, K., Legge, A., Fenn, M. E., Schilling, S., Fraczek, W., and Alexander, D.: Ground-level air pollution changes during a boreal wildland mega-fire, Sci Total Environ, 572, 755-769, 10.1016/j.scitotenv.2016.07.052, 2016.

Kelly, J. T., and Wexler, A. S.: Thermodynamics of carbonates and hydrates related to heterogeneous reactions involving mineral aerosol, Journal of Geophysical Research-Atmospheres, 110, 10.1029/2004jd005583, 2005.

Sullivan, R. C., Moore, M. J. K., Petters, M. D., Kreidenweis, S. M., Roberts, G. C., and Prather, K. A.: Effect of chemical mixing state on the hygroscopicity and cloud nucleation properties of calcium mineral dust particles, Atmos Chem Phys, 9, 3303-3316, 10.5194/acp-9-3303-2009, 2009.

20. *Page 9, Line 29-31, the result was obtained by the authors' lab experiment? If no, please add data prove or references.*

**Reply:** We consented to the reviewer's comments. The over discussion about chemical component here was removed to avoid misleading.

**Reply:** We agree with the reviewer's suggestions, corresponding lab experiment or composition observation data would enhance the credibility of the conclusions. We will revise the expression and concise the discussion about specific chemical reactions. We will revise the manuscript as "In general, the deliquescence capacity of natural dust particles are weak. When water soluble compositions were coated on the surface of the particles, their morphology would be more sensitive to environmental relative humidity. Numbers of studies have proved that the morphology change of coarse mode particles was highly affected by vapor content in the air, as Figure 14 suggests. A remarkable fact was that the depolarization ratio of coarse particles showed a significant negative correlation with RH. Take 3 μm particles for example, the DR value was 0.3 in dry air conditions (RH = 10%), and be as low as 0.15 when the humidity approaches 100% (50% decrease); DR for 5 μm particles showed a similar negative correlation (43% decrease). This kind of decrease in the depolarization ratio reflects the spheroidization of the dust particles. Kelly and Wexler (2005) found that crystalline hydrates of $Ca(NO3)_2$ and $CaCl_2$ formed at 9% and 0.6% RH as the critical values, and the corresponding deliquesce points were 50% and 28% RH, respectively. In mid-latitude regions, the typical ambient relative humidity ranges over 10% – 90%, typical calcium salts coating the outer layers of mineral dust aerosols are generally in the dissolution state. In a humid atmosphere, given that water film commonly acted as a solute for soluble component in atmosphere, inorganic salt crust on dust surface would have a potentially high impact on morphology change of mineral dust, which can be reflect by depolarization ratio decrease. Nevertheless, the theoretical simulation of Ishimoto et al. (2010) showed that a change in refractive index could also affect the DR value, the simulation results showed that the depolarization ratio showed a levelling off tendency at 0.31 ± 0.02 for the coarse modal particles (Figure 2), variations in the particle's refractive index can only explain limited depolarization variability (6%)."

[Figure]

Figure 2. Theoretical calculation of depolarization ratio (at 120 backward direction) as a function of particle size for different refractive index.

**Minor Comments:**

1. *Page 5, Line 11-13, "The growth of fine mode particles mainly stems from high anthropogenic emissions during heating seasons." confused with "The distinct growth period of fine mode particles was usually induced by an increase in relative humidity (i.e., RH)". Please afresh write the two sentences.*

   **Reply:** The sentences will be rewrite in the revised manuscript, as "In winter time, the overall concentration of fine particles was higher which can be attribute to high emissions for heating purpose and specific meteorology circumstance (Wang et al., 2015). The distinct growth period of fine mode particles was generally induced by an increase in relative humidity for all seasons"

   Wang, Q., Sun, Y., Jiang, Q., Du, W., Sun, C., Fu, P., and Wang, Z.: Chemical composition of aerosol particles and light extinction apportionment before and during the heating season in Beijing, China, Journal of Geophysical Research: Atmospheres, 120, 12708-12722, 10.1002/2015jd023871, 2015.

2. *Page 6, Line 31, what is JJ? Please explain it.*

   **Reply:** The "JJ" has been explained as June and July in the revised manuscript.

3. *Page 7, Line 3,4, NOx should be $NO_x$.*

   **Reply:** Thanks for pointing out the mistake, and the "NOx" has been revised as "$NO_x$" in the manuscript.

4. *Page 8, Line 6, please clearly point out "dominant mechanisms". What were dominant mechanisms?*

   **Reply:** I will delete the "dominant mechanisms" since it is not discussed in the context.

5. *Page 8, Line 10-11, the authors said the first processes (22st -23rd, Dec. 2015) and the third process (9th, Apr. 2016). But in Page 5, Line 17-18, the authors said "A" (December 22nd to 23rd, 2015), "B" (March 3rd to 4th, 2016) and "C" (April 9th, 2016), respectively. Please unite.*

   **Reply:** We will change the date in the manuscript to a uniform format.

6. *Page 8, Line 15, please added data descriptions or references for "primary anthropogenic pollution emissions in this period were also lower compared to the winter time".*

   **Reply:** We revised the sentence as suggested, as follows: " Considering the heating season was now over for a month, $PM_{2.5}$ that associated with high burning of fossil fuels for heating purpose was relatively weak (Wang et al., 2015). "

   Wang, Q., Sun, Y., Jiang, Q., Du, W., Sun, C., Fu, P., and Wang, Z.: Chemical composition of aerosol particles and light extinction apportionment before and during the heating season in Beijing, China, Journal of Geophysical Research: Atmospheres, 120, 12708-12722, 10.1002/2015jd023871, 2015.

7. *Page 9, Line 23-25, please added references.*

   **Reply:** We will cite Tao et al., (2013) to support relevant content, and change the manuscript as "One of the reasons was that $PM_{2.5}$ was reported to be mainly composed of anthropogenic pollutants

and only a small amount of mineral dust matter (Tao et al., 2013)."

Tao, J., Zhang, L., Engling, G., Zhang, R., Yang, Y., Cao, J., Zhu, C., Wang, Q., and Luo, L.: Chemical composition of PM$_{2.5}$ in an urban environment in Chengdu, China: Importance of springtime dust storms and biomass burning, Atmospheric Research, 122, 270-283, 10.1016/j.atmosres.2012.11.004, 2013.

8. *Figure 7f, the date is 04/10/2016, not 04/09/2016?*

**Reply:** The date on Figure 7f will be changed to "04/09/2016".

[Figure]

Figure 7: Backward trajectories from Beijing calculated by the FLEXPART dispersion model for (a) the anthropogenic pollution case, (b) the mixed pollution period, (c) the pure mineral dust dominant case. Variation in the standard depolarization ratio as a function of particle size for corresponding episodes: (d), (e), and (f). Emission information for FLEXPART is explained in section 2.2, while information about the footprint and trajectory analysis is presented in the introduction.

9. *Page 9, Line 24-25, please added references.*

**Reply:** This sentence will be changed as "One of the reasons was that PM$_{2.5}$ was reported to be mainly composed of anthropogenic pollutants and only a small amount of mineral dust matter (Tao et al., 2013)."

Tao, J., Zhang, L., Engling, G., Zhang, R., Yang, Y., Cao, J., Zhu, C., Wang, Q., and Luo, L.: Chemical composition of PM$_{2.5}$ in an urban environment in Chengdu, China: Importance of springtime dust storms and biomass burning, Atmospheric Research, 122, 270-283, 10.1016/j.atmosres.2012.11.004, 2013.

---

## Author Response (AR2)

**Reply to the comments of anonymous reviewer #1 on manuscript entitled "Variability of depolarization of aerosol particles in Beijing mega city: implication in interaction between anthropogenic pollutants and mineral dust particles"**

5    The authors appreciate very much for reviewing our manuscript and your insight comments. As suggested, we carefully revised the manuscript thoroughly according to the valuable advices, as well as proof-read the manuscript to minimize typographical, grammatical, and bibliographical errors.

**General Comments:**

10    *Tian et al. discuss aerosol particle polarization properties from November 29, 2015 to July 29, 2016 at Beijing. The dataset presented is interesting, particularly in showing particle polarization in anthropogenic pollution, mixed type pollution and pure Asian dust cases. Using an optical particle counter equipped with a polarization detection module is a new method to study mixing processes. However, the discussion in the current draft is insufficient for these data and results: a number of*
15    *results/conclusions are not obtained from lab experiment or data prove, just derivate from published paper. I believe the article therefore should add data prove or lab experiment, and require major revisions.*

**Reply:** We fully agree with the reviewer's suggestions that simultaneous measurement of chemical compositions of particles and lab experiment evaluations will benefit the understanding of the aerosol's
20    physical changes during mixed pollution. The change in aerosol's depolarization ratio can only represent that overall characteristics of the surface of dust aerosols, not the chemical reactions on it. As the reviewer suggested, we add more observational and experimental results in laboratory to clarify the chemical process happened during mix pollution, and the depolarization ratio of particles with meteorological factors, $PM_{2.5}$ and $PM_{10}$ concentrations, and aerosol chemical compositions are carefully discussed. To
25    respond to the reviewer's major concerns, we made thoroughly revisions and corrections according to all the insight comments of the reviewers. Besides, more crucial/supporting information and analysis will be added in the revised manuscript, as follows:

(1) A detailed description and specification of calibration system of POPC and calibration results in this study will be added in the revised manuscript.

30    (2) Comparison between observation and the simulation result with T-matrix method on randomly-oriented Voronoi aggregation and elongated ellipsoid particles will be added.

(3) A comparison result of measurement of size-resolved number concentration of particles between POPC and KC52 optical particle counter will be added in the revised manuscript.

(4) Comparison of observation results between POPC measurement and LIDAR observation during
35    several dust events in Beijing will be discussed. The cross validation on the dust events provide more direct evidence of variation in the depolarization properties of dust particles.

Besides, to make the conclusions more pertinent to this study, we revised the expression and concise the discussion about specific chemical composition and reaction. Our conclusions were useful for understanding the long-term depolarization property of aerosols in East Asia, interaction between anthropogenic pollutions and mineral dust in East Asia and their impacts on local air quality, and the sensitive relation between particle morphology and meteorological parameter. This long-term study in Beijing motivate revisit on decades of Lidar data in dust-pollution interactions, and it indicates the necessity of a reliable optical model of internally mixed polluted dust for a detailed analysis of polarization remote sensing observations.

The initial purpose of this observational study is to investigate the long-term morphology properties of particles in Beijing because the North China region has long experienced seasonal haze and dust events due to the special geographical location and industrial development stage. As far as we know, due to limit of observation techniques, there was no studies reported on long-term variability of morphological characteristics of aerosols in north China till now. Alternatively, depolarization ratio of single particle is an applicable parameter indicating of general irregularity of aerosol particles. For this reason, an eight-month field measurement was performed at an urban site in Beijing megacity, we detected single particle depolarization ratio properties using a new developed beach-top instrument. Take advantage of meteorology/pollution data and trajectory analysis during the same period, the relationship between morphology change of particles and meteorological factors, especially water content in the atmosphere, was investigate. As pointed out, electro-microscopy inspection and composition results would provide direct evidence of chemo-physical changes of the particles, we will add more supporting information in the revised manuscript. The knotty point is that, single particle-based laboratory inspections need passive samples with variable time scales of 1 min to several days, it is difficult to have enough samplings to represent the long-term variability of aerosols statistically. We will follow reviewer's advice to conduct single-particle based electro-microscopic study in our next study. Chemical compositions of particles is helpful for understanding the possible chemical reaction during observation period. Unfortunately, we didn't have concurrent measurement during the experiment.

According to long-term real-time measurements of aerosol particle composition in Beijing in previous study, the seasonal variations of secondary inorganic aerosol (SIA, i.e., sulfate C nitrate C ammonium) concentrations were not significant, contributions of SIA were observed in summer (57 – 61 %) and in winter (43 – 46 %) in fine mode aerosols (Sun et al., 2015). Although we did not have *in situ* measurement in this study in Beijing, a high $PM_{2.5}/PM_{10}$ ratio still indicates that the components of soluble inorganic salts was greatly possible to be relatively high, and in the case of high humidity. Therefore, morphology and corresponding depolarization ratio of particles should be affected. As expected, I did find a similar correlation between the ratio of $PM_{2.5}/PM_{10}$ and the depolarization ratio of coarse-mode aerosols, proving the great possibility that the inorganic salts might be trapped by large particles, or the occurrence of reaction between reactive gas and alkaline dust particles. At present, depolarization ratio is deemed as the most applicable method to rapidly or simultaneously distinguish types or mixing state of pollutions. Such characteristic are widely used by satellite-borne remote sensing and ground-based Lidar observations (Burton et al., 2012; Shimizu et al., 2004a; Shimizu et al., 2004b). In this study, combining analysis with LIDAR measurement, we hope to find an alternative and relatively simple indicator, such

as the ratio of $PM_{2.5}/PM_{10}$, to estimate the possibility of morphology change of particles in the coarse-mode. In the manuscript, we revised the expression and weaken the discussion about specific chemical composition and reaction. In our next research, we would like to follow reviewer's suggestions to perform comprehensive measurements on the dependence of both morphology properties and chemical information of particles in different mixing state.

Sun, Y. L., Wang, Z. F., Du, W., Zhang, Q., Wang, Q. Q., Fu, P. Q., Pan, X. L., Li, J., Jayne, J., and Worsnop, D. R.: Long-term real-time measurements of aerosol particle composition in Beijing, China: seasonal variations, meteorological effects, and source analysis, Atmos Chem Phys, 15, 10149-10165, 10.5194/acp-15-10149-2015, 2015.

Burton, S. P., Ferrare, R. A., Hostetler, C. A., Hair, J. W., Rogers, R. R., Obland, M. D., Butler, C. F., Cook, A. L., Harper, D. B., and Froyd, K. D.: Aerosol classification using airborne High Spectral Resolution Lidar measurements – methodology and examples, Atmospheric Measurement Techniques, 5, 73-98, 10.5194/amt-5-73-2012, 2012.

Shimizu, A., Sugimoto, N., Matsui, I., Arao, K., Uno, I., Murayama, T., Kagawa, N., Aoki, K., Uchiyama, A., and Yamazaki, A.: Continuous observations of Asian dust and other aerosols by polarization lidars in China and Japan during ACE‐Asia, Journal of Geophysical Research: Atmospheres, 109, 2004a.

Shimizu, A., Sugimoto, N., Matsui, I., Arao, K., Uno, I., Murayama, T., Kagawa, N., Aoki, K., Uchiyama, A., and Yamazaki, A.: Continuous observations of Asian dust and other aerosols by polarization lidars in China and Japan during ACE-Asia, Journal of Geophysical Research-Atmospheres, 109, 10.1029/2002jd003253, 2004b.

**Specific Comments.**

1. *Page 2, Line 2-3, only consumption of fossil fuel lead serious air pollution in East Asia? How about vehicle exhaust?*

   **Reply:** We agree to the reviewer's comments. On-road vehicle exhaust plays an important role in formation of urban air pollution. We revised the expression in the manuscript as following, "Rapid economic growth and urbanization processes in East Asia has caused serious air pollution in the past decades owing to the substantial consumption of fossil fuel. Anthropogenic emissions (industrial, traffic, residential etc.) emits substantial amount of pollutant gases ($SO_2$, $NO_2$, $NH_3$, VOCs etc.) and primary aerosols (Akimoto, 2003;Akimoto et al., 2006;Kurokawa et al., 2013;Li et al., 2017), resulting in formation of $PM_{2.5}$."

Akimoto, H.: Global air quality and pollution, Science, 302, 1716-1719, 10.1126/science.1092666, 2003.

Akimoto, H., Ohara, T., Kurokawa, J.-i., and Horii, N.: Verification of energy consumption in China during 1996–2003 by using satellite observational data, Atmospheric Environment, 40, 7663-7667, 10.1016/j.atmosenv.2006.07.052, 2006.

Kurokawa, J., Ohara, T., Morikawa, T., Hanayama, S., Janssens-Maenhout, G., Fukui, T., Kawashima, K., and Akimoto, H.: Emissions of air pollutants and greenhouse gases over Asian regions during 2000–2008: Regional Emission inventory in ASia (REAS) version 2, Atmos Chem Phys, 13, 11019-11058, 10.5194/acp-13-11019-2013, 2013.

Li, M., Zhang, Q., Kurokawa, J.-i., Woo, J.-H., He, K., Lu, Z., Ohara, T., Song, Y., Streets, D. G., Carmichael, G. R., Cheng, Y., Hong, C., Huo, H., Jiang, X., Kang, S., Liu, F., Su, H., and Zheng, B.: MIX: a mosaic Asian anthropogenic emission inventory under the international collaboration framework of the MICS-Asia and HTAP, Atmos Chem Phys, 17, 935-963, 10.5194/acp-17-935-2017, 2017.

2. *Page 2, Line 6, mineral dust particles only originated from natural sources? In the manuscript 3.4 part, the authors said emissions in famous mining areas in China can result in a huge amount of dust.*

**Reply:** The statement in section 3.4 made a misleading. What we want to express is that mining areas produce huge amount of pollutant gases $SO_2$, $CO$, as well as $NO_x$ related to vehicle exhaust in North China. In case C, there was a convergent air mass from natural dust sources (desert and Gobi) and mining areas in Beijing, according to FLEXPART trajectory analysis. It lead to a huge amount of dust mixed with pollutant-rich gaseous atmosphere in the Beijing area. We will revise the expression in the revised manuscript in section 3.5.1. As the reviewer pointed out, mineral dust particles in city environment might also be emitted from on-road vehicle and construction dust. These fugitive dust usually have a large diameter and a short lifetime due to rapid gravity settlement velocity. The sentence has been revised as "Mineral dust particles in the atmosphere also have a detrimental impact on air quality and human health such as reducing visibility, increase respiratory morbidity."

3. *Page 3, Line 3, the authors should add more descriptions about depolarization of aerosol particles.*

**Reply:** We will add more descriptions about depolarization of particles. Volume depolarization ratio of particle groups is widely used in LIDAR detection to distinguish the types of aerosol particles. The volume depolarization information of target air parcel just represents an overall depolarization. In this study, we use a new instrument to measure real-time variations in single particle depolarization ratio. This measurement can avoid the influences from the presence of substantial amount of small spherical particles. As mentioned, direction of polarization of scattering light was identical with the incident light for spherical particle; however, for the non-spherical particles, the direction of polarization of light will be deviated. According to such phenomenon, mineral dust and anthropogenic pollution dominant particles can be distinguished. Sugimoto et al. (2015) found that backscattering depolarization ratio in polluted dust was smaller compared to pure Asia dust, Pan et al. (2017) reported the vital role in observed depolarization ratio decrease that calcium nitrate played via coating process and hygroscopic growth on the surface of dust particles. In a word, single particle-based depolarization ratio is applicable for quantitatively investigating the evolution of the mixing of dust particles during their transport. As suggested by the reviewer, we will add and clarify the relevant explanation about depolarization ratio in the revised manuscript.

Sugimoto, N., Nishizawa, T., Shimizu, A., Matsui, I., and Kobayashi, H.: Detection of internally mixed Asian dust with air pollution aerosols using a polarization optical particle counter and a polarization-sensitive two-wavelength lidar, Journal of Quantitative Spectroscopy & Radiative Transfer, 150, 107-113, 10.1016/j.jqsrt.2014.08.003, 2015.

Pan, X., Uno, I., Wang, Z., Nishizawa, T., Sugimoto, N., Yamamoto, S., Kobayashi, H., Sun, Y., Fu, P., Tang, X., and Wang, Z.: Real-time observational evidence of changing Asian dust morphology with the mixing of heavy anthropogenic pollution, Sci Rep, 7, 335, 10.1038/s41598-017-00444-w, 2017.

4. *Page 3, Line 4-5, the authors said studies on the polarization characteristics are necessary, how about depolarization characteristics?*

**Reply:** To avoid misleading, we will use depolarization ratio of particle in the whole context. Actually, identical laser hit particles and producing different polarized light signals. The different is caused by particle morphology and best described by the ratio of "depolarization". I will make corresponding corrections in revised manuscript.

5. *Page 3, Line 10-14, why key pollutants and meteorological data were obtained from two different sites? The key pollutants and meteorological data are the same in the two different sites?*

**Reply:** We select the nearest state-controlled environmental monitoring station and weather station to the observation experiment site to support this study. As known, the horizontal scale of an air mass is 10 kilometers to hundreds of kilometers, the synoptic condition of the air mass (pollutant concentration, humidity and temperature) in city scale are normally the similar (as shown in Figure 1), though differences existed due to possible local sources.

[Figure]

**Figure 1. Time serious of particle mass concentration of PM$_{2.5}$ and PM$_{10}$, and meteorology factors (relative humidity and temperature) in different site in Beijing area.**

6. *Page 4, Line 28-29, why 150 m above ground level was chose? Why not is 100m or 200m? Why the simulation time of the backward trajectory is 5 days, not 3 days or 2 days?*

**Reply:** We performed backward trajectory simulation starting from height of 150 m, because our observation is on a top of two-story building (~10 meter high above the ground), and urban canopy of Beijing city was estimated to be less than 200 m. Considering that we use trajectories to indicate synoptic transport of dust aerosols, a starting height of 150 m is chosen. As a matter of fact, the trajectories starting from 150 m and 200 m are similar, and did not change our major verdict. A weather process is generally last for 3-7 days. We choose 5 days simulation because it can better reflect the long-distance transmission of particles. 2 or 3 days simulation may only reflect the local impact, but the typically continuous dust process has a great probability to be influenced by long-distance transmission in North China.

7. *Page 5, Line 9-11, What about the sources of particle size in 1 μm – 4 μm? A number of published papers said that PM$_{2.5}$ also mostly derived from anthropogenic sources. The reference of Kurosaki and Mikami, (2003) is a bit old.*

**Reply:** The particle with aerodynamic diameter less than 2.5 μm is defined as fine mode, and the particle size between 2.5 – 10 μm is defined as coarse mode. As mentioned, the particles between 1 – 4 μm consisted of both anthropogenic pollutants such as sulfate, nitrate and organic matters and mineral dust particle from both nature and anthropogenic sources. Numbers of previous studies indicate that the mode diameter of dust particle in North China is normally about 5 μm. It was consistent with our measurement of size distribution which showed a broad peak between 4 – 8 μm. We would like to cite Lue et al. (2010) to support our conclusion in the revised manuscriptLue, Y. L., Liu, L. Y., Hu, X., Wang, L., Guo, L. L., Gao, S. Y., Zhang, X. X., Tang, Y., Qu, Z. Q., Cao, H. W., Jia, Z. J., Xu, H. Y., and Yang, Y. Y.: Characteristics and provenance of dustfall during an unusual floating dust event, Atmospheric Environment, 44, 3477-3484, 10.1016/j.atmosenv.2010.06.027, 2010.

8. *Page 5, Line 24-26, the depolarization ratios of fine mode particles in summer are not lower than any other season? Why the authors emphasized depolarization ratios of particles with an optical diameter of 2 μm are lower than any other season?*

**Reply:** Sorry for misleading here. The depolarization ratios of all mode particles in summer were lower than other seasons, as shown in Figure 2b-c and Figure 5 in the manuscript. This seasonal variability was more obvious for fine mode aerosols. It was because in summer, high relative humidity could lead to absorption and deliquesce process happened on soluble component in aerosols. Depolarization ratio of 2 μm particles was separately mentioned as an example. We will clarify this point in the revised manuscript as "In general, the $\delta$ value of particles in urban Beijing had prominent seasonal variability with a summer-low and spring-high pattern due to the different composition, origins of aerosols and atmospheric meteorology at the site. The averaged $\delta$ values of particles in both fine and coarse mode was highest in March 2016 (0.26) and lowest in July 2016 (0.19). This seasonal variability was much obvious in fine mode particles……"

9. *Page 6, Line 4-7, wind speed, biomass burning or other factors have no impact on high depolarization ratio? Only transport lead high depolarization ratio?*

**Reply:** We consent to the reviewer's comment that depolarization properties of different aerosol types are different. For instance, the soot particles from biomass burning normally have complex chain-like morphology which was distinct from spherical particles. In this study, the depolarization

ratio actually reflect the type of particle: mineral dust, and anthropogenic pollutant or sea salt aerosols (Kobayashi et al., 2014). We will revise the interpretation in the manuscript as "For mineral dust-dominant episodes, the air mass mainly originated from large areas in western Mongolia and Gobi Desert, and the footprint pattern represented comparatively large dust loading in the atmosphere in spring. While for anthropogenic pollution-dominant period, air mass past through Beijing-Tianjin-Hebei region where anthropogenic emission was significantly strong. To note that, the RH along the trajectories was 13.9 % averagely during dust-dominant dominate cases (mostly in spring-time) and 87.6 % in anthropogenic pollution-dominant cases (mostly in winter-time). It means that origin of aerosol particles and their interaction with water vapor as well as consecutively heterogeneous reactions can lead to pronounced morphological changes of particles."

Kobayashi, H., Hayashi, M., Shiraishi, K., Nakura, Y., Enomoto, T., Miura, K., Takahashi, H., Igarashi, Y., Naoe, H., Kaneyasu, N., Nishizawa, T., and Sugimoto, N.: Development of a polarization optical particle counter capable of aerosol type classification, Atmospheric Environment, 97, 486-492, 10.1016/j.atmosenv.2014.05.006, 2014.

10. *Page 6, Line 7-10, have the authors ruled out other factors (such as wind speed, biomass burning)? These factors also can lead high $PM_{2.5}$.*

**Reply:** Yes, as pointed out by the reviewer, factors including wind speed and biomass burning also can lead high $PM_{2.5}$. Previous studies of Wang et al. (2013) and Chang and Zhan (2017) have demonstrated the necessary effect of weak wind speed on haze formation. Chen et al. (2017) and Cheng et al. (2014) have also demonstrated the importance of biomass burning high $PM_{2.5}$ levels by observing facts. It reflected that the biomass combustion and specific synoptic situation was of great importance in variation of $PM_{2.5}$. In this study, we forces on the depolarization properties in ambient air in Beijing area, and the trajectory analysis was to explain the seasonal change of depolarization ratio. In our next study, we would like to quantify the relationship between biomass burning and $PM_{2.5}$ levels, and investigate the change and effect on depolarization ratio of ambient air.

Wang, Y., Yao, L., Wang, L., Liu, Z., Ji, D., Tang, G., Zhang, J., Sun, Y., Hu, B., and Xin, J.: Mechanism for the formation of the January 2013 heavy haze pollution episode over central and eastern China, Science China Earth Sciences, 57, 14-25, 10.1007/s11430-013-4773-4, 2013.

Chang, W., and Zhan, J.: The association of weather patterns with haze episodes: Recognition by PM2.5 oriented circulation classification applied in Xiamen, Southeastern China, Atmospheric Research, 197, 425-436, 10.1016/j.atmosres.2017.07.024, 2017b.

Chen, J., Li, C., Ristovski, Z., Milic, A., Gu, Y., Islam, M. S., Wang, S., Hao, J., Zhang, H., He, C., Guo, H., Fu, H., Miljevic, B., Morawska, L., Thai, P., Lam, Y. F., Pereira, G., Ding, A., Huang, X., and Dumka, U. C.: A review of biomass burning: Emissions and impacts on air quality, health and climate in China, Sci Total Environ, 579, 1000-1034, 10.1016/j.scitotenv.2016.11.025, 2017.

Cheng, Z., Wang, S., Fu, X., Watson, J. G., Jiang, J., Fu, Q., Chen, C., Xu, B., Yu, J., Chow, J. C., and Hao, J.: Impact of biomass burning on haze pollution in the Yangtze River delta, China: a case study in summer 2011, Atmos Chem Phys, 14, 4573-4585, 10.5194/acp-14-4573-2014, 2014.

11. *Page 6, Line 30-31, the authors said "Mixed type pollution was also necessary for substandard days in Jan. (44%), Feb. (50%), and Apr. (35%)". But in Page 5, Line 17-18, the authors said mixed type pollution is "B" (March 3rd to 4th, 2016). And please clearly describe how to decide mixed type pollution.*

**Reply:** In this study, mixed type pollution means coexistence of anthropogenic pollutants and dust particles in the atmosphere. We will make it clear in the revised manuscript. To better reflect the influence of dust-related heterogeneous processes on pollution formation, we changed this part into the comparison of $\delta$ value in substandard days and clean days as shown in Figure 5 and section 3.4 in the revised manuscript. We accept suggestions from the reviewer and will further explain and discuss the reasons and vital influence of mixed pollution in the revised manuscript.

*12. Page 7, Line 7-9, the authors obtained the conclusion is based on the report in published papers, not obtained from the authors' observation or lab experiment. So possibly the conclusion didn't confirm.*

**Reply:** We agree the suggestion of reviewer and will carefully make discussion and conclusion in revised manuscript as "This implies that pollution days in North China was great possible to induce internal mixing of dust and pollutant, especially in high humid atmospheric environment, and dust-related heterogeneous processes on the dust surface can aggravate the deterioration of air quality as a feedback."

*13. Page 7, Line 25-26, why the authors confirm that increasing $PM_{2.5}$ level in case A was caused by weak diffusion conditions and high relative humidity? How about other aspects, such as emissions?*

**Reply:** The heating season in north China is from mid-November to mid-March of the following year. The pollutant emission in Beijing area in this period remained at a high level compared with any other seasons (Wang et al., 2015). Haze pollution can essentially be attributed to two aspects: pollutant emissions into the lower atmosphere and favorable meteorological conditions. In particular, meteorological conditions control the occurrence of haze pollution (Zhang et al., 2016; Wang et al., 2013; Jacob and Winner, 2009; Ilten and Selici, 2008; Chang and Zhan, 2017). High emission of primary pollutant precursors in winter time combine with stable weather conditions (weak diffusion conditions) and favorable conditions for liquid phase reaction of pollutant (high humid), resulting in the formation of secondary pollutants $PM_{2.5}$. And the clarity of the pollution process is usually accompanied by a strong cold air which from Siberia spills south into East Asia. We will make the reasons for high $PM_{2.5}$ clearer in the manuscript.

Wang, Q., Sun, Y., Jiang, Q., Du, W., Sun, C., Fu, P., and Wang, Z.: Chemical composition of aerosol particles and light extinction apportionment before and during the heating season in Beijing, China, Journal of Geophysical Research: Atmospheres, 120, 12708-12722, 10.1002/2015jd023871, 2015.

Zhang, Z., Zhang, X., Gong, D., Kim, S. J., Mao, R., and Zhao, X.: Possible influence of atmospheric circulations on winter haze pollution in the Beijing–Tianjin–Hebei region, northern China, Atmos Chem Phys, 16, 561-571, 10.5194/acp-16-561-2016, 2016.

Wang, Y., Yao, L., Wang, L., Liu, Z., Ji, D., Tang, G., Zhang, J., Sun, Y., Hu, B., and Xin, J.: Mechanism for the formation of the January 2013 heavy haze pollution episode over central and eastern China, Science China Earth Sciences, 57, 14-25, 10.1007/s11430-013-4773-4, 2013.

Jacob, D. J., and Winner, D. A.: Effect of climate change on air quality, Atmospheric Environment, 43, 51-63, 10.1016/j.atmosenv.2008.09.051, 2009.

Ilten, N., and Selici, A. T.: Investigating the impacts of some meteorological parameters on air pollution in Balikesir, Turkey, Environ Monit Assess, 140, 267-277, 10.1007/s10661-007-9865-1, 2008.

Chang, W., and Zhan, J.: The association of weather patterns with haze episodes: Recognition by PM2.5 oriented circulation classification applied in Xiamen, Southeastern China, Atmospheric Research, 197, 425-436, 10.1016/j.atmosres.2017.07.024, 2017.

*14. Page 7, Line 29-33, Fig, 7b didn't present height of air parcel, and in the paper, the authors also didn't present descriptions about height of air parcel in the draft. Why the authors can get conclusion that emissions in mining areas can lead a huge amount of dust? And in the paper, the descriptions about emissions in mining areas in March 3 to 4 are also not given.*

**Reply:** As suggested, we will add detailed simulation information in section 2.2 and section 3.5. In the case C, the air mass was a convergence of deviating northwest and south streams. The eastward-conveying air mass originated from the Taklimakan Desert in Sinkiang and channeled the Tengger Desert in Inner Mongolia, which were major source of dust in east Asia (Fu et al., 2012; Chen et al., 2013; Chen et al., 2017; Zhang et al., 2018); the south branch of the air mass past central Henan Province and southern Hebei Province where emission of gaseous pollutants from industries, residential and transport sources were significant (Huang et al., 2012;Zhao et al., 2012;Wang et al., 2014). What we want to express here was the eastward transport air mass may lead to a huge amount of dust mixed with pollutant-rich gaseous atmosphere in the Beijing area. We will make a more clear and accurate revision in the section 3.5.3 in the manuscript.

Fu, P., Zhong, S., Huang, J., and Song, G.: An observational study of aerosol and turbulence properties during dust storms in northwest China, Journal of Geophysical Research: Atmospheres, 117, n/a-n/a, 10.1029/2011jd016696, 2012.

Chen, S., Huang, J., Zhao, C., Qian, Y., Leung, L. R., and Yang, B.: Modeling the transport and radiative forcing of Taklimakan dust over the Tibetan Plateau: A case study in the summer of 2006, Journal of Geophysical Research: Atmospheres, 118, 797-812, 10.1002/jgrd.50122, 2013.

Chen, S., Huang, J., Kang, L., Wang, H., Ma, X., He, Y., Yuan, T., Yang, B., Huang, Z., and Zhang, G.: Emission, transport, and radiative effects of mineral dust from the Taklimakan and Gobi deserts: comparison of measurements and model results, Atmos Chem Phys, 17, 2401-2421, 10.5194/acp-17-2401-2017, 2017.

Zhang, X.-X., Sharratt, B., Liu, L.-Y., Wang, Z.-F., Pan, X.-L., Lei, J.-Q., Wu, S.-X., Huang, S.-Y., Guo, Y.-H., Li, J., Tang, X., Yang, T., Tian, Y., Chen, X.-S., Hao, J.-Q., Zheng, H.-T., Yang, Y.-Y., and Lyu, Y.-L.: East Asian dust storm in May 2017: observations, modelling and its influence on Asia-Pacific region, Atmospheric Chemistry and Physics Discussions, 1-31, 10.5194/acp-2018-205, 2018.

Huang, X., Song, Y., Li, M., Li, J., Huo, Q., Cai, X., Zhu, T., Hu, M., and Zhang, H.: A high-resolution ammonia emission inventory in China, Global Biogeochemical Cycles, 26, n/a-n/a, 10.1029/2011gb004161, 2012.

Zhao, B., Wang, P., Ma, J. Z., Zhu, S., Pozzer, A., and Li, W.: A high-resolution emission inventory of primary pollutants for the Huabei region, China, Atmos Chem Phys, 12, 481-501, 10.5194/acp-12-481-2012, 2012.

Wang, L. T., Wei, Z., Yang, J., Zhang, Y., Zhang, F. F., Su, J., Meng, C. C., and Zhang, Q.: The 2013 severe haze over southern Hebei, China: model evaluation, source apportionment, and policy implications, Atmos Chem Phys, 14, 3151-3173, 10.5194/acp-14-3151-2014, 2014.

*15. Page 8, Line 6-9, please connected the cited part of Itahashi et al. (2010) with your study.*

**Reply:** In the original manuscript version, we cite the work of Itahashi to explain the descending of dust plume on March 5 in case C. Considering that the discussion of meteorological factors is not the focus of this study, we removed this part in the revised manuscript. For better discuss about the internal/external mixing state of dust particle, and compare the two depolarization ratio detecting

instruments which is Lidar and POPC, we will add more observation result for Lidar at the same period in section 3.52 and section 3.5.3.

16. *Page 8, Line 15, please give data prove or corresponding references.*

**Reply:** Considering the heating season was now over for a month, $PM_{2.5}$ that associated with high burning of fossil fuels for heating purpose was relatively weak, we would like to cite Wang et al., (2015) in the manuscript,

Wang, Q., Sun, Y., Jiang, Q., Du, W., Sun, C., Fu, P., and Wang, Z.: Chemical composition of aerosol particles and light extinction apportionment before and during the heating season in Beijing, China, Journal of Geophysical Research: Atmospheres, 120, 12708-12722, 10.1002/2015jd023871, 2015.

17. *Page 8, Line 16-17, how does the author judge no anthropogenic pollution emissions in case C, and obtain a pure dust process? In the paper, there are no data prove.*

**Reply:** For easier and better understanding of the manuscript, we redefined the time of the two dust case, that was case B on April 9-10 and case C on March 4-6, 2016. We agree with the reviewer that we cannot describe case B as completely free of anthropogenic pollution due to lacking of chemical composition information. In the period, the daily $PM_{10}$ was 273.6 $\mu g/m^3$ and $PM_{2.5}$ account for 33.4% in $PM_{10}$ on April 9. POPC analysis (Figure 6e in the manuscript) showed that particles in this process was concentrated in the particle size range larger than 3 $\mu m$, in which that contribution secondary formed aerosol particles was less important. During case B, air mass originated from Gobi desert in Mongolia and transport rapidly by a strong wind. Recent filter-based observation (Pan et al., 2017) showed that the water-soluble compounds in coarse mode was insignificant, compared to crustal matters, during dust event. To avoid misleading, we will defined case B as a typical dust-dominant period. We will make a more comprehensive interpretation of this conclusion in the revised manuscript according to the comments of reviewers.

Pan, X., Uno, I., Wang, Z., Nishizawa, T., Sugimoto, N., Yamamoto, S., Kobayashi, H., Sun, Y., Fu, P., Tang, X., and Wang, Z.: Real-time observational evidence of changing Asian dust morphology with the mixing of heavy anthropogenic pollution, Sci Rep, 7, 335, 10.1038/s41598-017-00444-w, 2017.

18. *Page 9, Line 21-22, why the authors use $PM_{2.5}/PM_{10}$ ratio? In 3.1 part, the authors said the fine mode (<1 μm), the coarse mode (4 μm – 8 μm). Why the authors didn't use $PM_{1.0}$, $PM_{4.0-8.0}$?*

**Reply:** Sorry for the misleading. The particles with aerodynamic diameter less than 2.5 $\mu m$ is defined as fine mode, and the particles with diameter between 2.5 and 10 $\mu m$ is defined as coarse mode. The widely adopted commercial online instruments normally measure the mass concentrations of $PM_{2.5}$ and $PM_{10}$. For POPC measurement, the mass concentrations of particles of $PM_{1.0}$ and $PM_{4.0-8.0}$ could be reasonably calculated on the basis of size distribution of particles. However, this measurement at the state-control observatories are very limited in China, which make it difficult to estimate the variability of aerosol types. Alternatively, we adopted the ratio $PM_{2.5}/PM_{10}$ to identify the sources of primary pollutant. A higher ratio was generally ascribed to anthropogenic related secondary particles (sulfate, nitrate etc.), and a lower ratio indicate significant contributions mainly re-suspended or fugitive mineral dust particles due to some mechanical processes (Chan et al., 2005; Pérez et al., 2008; Akyuz and Cabuk, 2009; Chan and Yao, 2008; Xu et al., 2017). In a typical dust

period, the higher the ratio was, the more possibility that anthropogenic pollution and dust particles were mixing, and consequentially morphological changes. We agree with the reviewer's concerns and will add the explanations for using the $PM_{2.5}/PM_{10}$ ratio in the manuscript.

5   Chan, C. Y., Xu, X. D., Li, Y. S., Wong, K. H., Ding, G. A., Chan, L. Y., and Cheng, X. H.: Characteristics of vertical profiles and sources of PM2.5, PM10 and carbonaceous species in Beijing, Atmospheric Environment, 39, 5113-5124, 10.1016/j.atmosenv.2005.05.009, 2005.

Pérez, N., Pey, J., Querol, X., Alastuey, A., López, J. M., and Viana, M.: Partitioning of major and trace components in PM10–PM2.5–PM1 at an urban site in Southern Europe, Atmospheric Environment, 42, 1677-1691, 10.1016/j.atmosenv.2007.11.034, 2008.

10  Akyuz, M., and Cabuk, H.: Meteorological variations of PM2.5/PM10 concentrations and particle-associated polycyclic aromatic hydrocarbons in the atmospheric environment of Zonguldak, Turkey, J Hazard Mater, 170, 13-21, 10.1016/j.jhazmat.2009.05.029, 2009.

Chan, C. K., and Yao, X.: Air pollution in mega cities in China, Atmospheric Environment, 42, 1-42, 10.1016/j.atmosenv.2007.09.003, 2008.

15  Xu, G., Jiao, L., Zhang, B., Zhao, S., Yuan, M., Gu, Y., Liu, J., and Tang, X.: Spatial and Temporal Variability of the PM2.5/PM10 Ratio in Wuhan, Central China, Aerosol and Air Quality Research, 17, 741-751, 10.4209/aaqr.2016.09.0406, 2017.

19. *Page 9, Line 25-26, generally we firstly presented some evidences, such as data prove or results in published paper, then derivate possible conclusion.*

20  **Reply:** We will cite more relevant reference to support our conclusion in the revised manuscript as "This negative relationship reflected the spheroidization of the dust particles as a result of hygroscopic properties of mineral dust aerosols. Polluted air generally contains abundant $HNO_3$ (Liang et al., 2007; Shi et al., 2014; Bytnerowicz et al., 2016). Li and Shao (2009) reported that mineral particles are mainly covered with coating including $Ca(NO_3)_2$, $Mg(NO_3)_2$, and $NaNO_3$ in

25  North China. Deeper interaction between alkaline mineral dust and reactive acidic gases and the trapping process of atmospheric secondary inorganic salt modified the hydrophilic state of dust aerosols."

Liang, Q., Jaeglé, L., Hudman, R. C., Turquety, S., Jacob, D. J., Avery, M. A., Browell, E. V., Sachse, G. W., Blake, D. R., Brune, W., Ren, X., Cohen, R. C., Dibb, J. E., Fried, A., Fuelberg, H., Porter,
30  M., Heikes, B. G., Huey, G., Singh, H. B., and Wennberg, P. O.: Summertime influence of Asian pollution in the free troposphere over North America, Journal of Geophysical Research, 112, 10.1029/2006jd007919, 2007.

Shi, Y., Chen, J., Hu, D., Wang, L., Yang, X., and Wang, X.: Airborne submicron particulate (PM1) pollution in Shanghai, China: chemical variability, formation/dissociation of associated semi-
35  volatile components and the impacts on visibility, Sci Total Environ, 473-474, 199-206, 10.1016/j.scitotenv.2013.12.024, 2014.

Bytnerowicz, A., Hsu, Y. M., Percy, K., Legge, A., Fenn, M. E., Schilling, S., Fraczek, W., and Alexander, D.: Ground-level air pollution changes during a boreal wildland mega-fire, Sci Total Environ, 572, 755-769, 10.1016/j.scitotenv.2016.07.052, 2016.

Li, W. J., and Shao, L. Y.: Observation of nitrate coatings on atmospheric mineral dust particles, Atmos Chem Phys, 9, 1863-1871, DOI 10.5194/acp-9-1863-2009, 2009.

20. *Page 9, Line 29-31, the result was obtained by the authors' lab experiment? If no, please add data prove or references.*

**Reply:** We consented to the reviewer's comments. The over discussion about chemical component here was removed to avoid misleading.

21. *Page 10, Line 3-20, please give corresponding data prove or lab experiment, then obtain conclusions.*

**Reply:** We agree with the reviewer's suggestions, corresponding lab experiment or composition observation data would enhance the credibility of the conclusions. We will revise the expression and concise the discussion about specific chemical reactions in section 3.6.

**Minor Comments:**

1. *Page 5, Line 11-13, "The growth of fine mode particles mainly stems from high anthropogenic emissions during heating seasons." confused with "The distinct growth period of fine mode particles was usually induced by an increase in relative humidity (i.e., RH)". Please afresh write the two sentences.*

**Reply:** The sentences will be rewrite in the revised manuscript, as "It can be seen in Figure 2a that volume size distribution generally had two size modes during the whole observation periods. The occurrence of fine mode (peak at ~1 μm) was accompanied with increase of RH.

2. *Page 6, Line 31, what is JJ? Please explain it.*

**Reply:** The "JJ" has been explained as June and July in the revised manuscript.

3. *Page 7, Line 3,4, NOx should be $NO_x$.*

**Reply:** Thanks for pointing out the mistake, and the "NOx" has been revised as "$NO_x$" in the manuscript.

4. *Page 8, Line 6, please clearly point out "dominant mechanisms". What were dominant mechanisms?*

**Reply:** I will delete the "dominant mechanisms" since it is not discussed in the context.

5. *Page 8, Line 10-11, the authors said the first processes (22st -23rd, Dec. 2015) and the third process (9th, Apr. 2016). But in Page 5, Line 17-18, the authors said "A" (December 22nd to 23rd, 2015), "B" (March 3rd to 4th, 2016) and "C" (April 9th, 2016), respectively. Please unite.*

**Reply:** We will change the date in the manuscript to a uniform format.

6. *Page 8, Line 15, please added data descriptions or references for "primary anthropogenic pollution emissions in this period were also lower compared to the winter time".*

**Reply:** We revised the sentence as suggested, as follows: " Considering the heating season was now over for a month, $PM_{2.5}$ that associated with high burning of fossil fuels for heating purpose was relatively weak (Wang et al., 2015). "

Wang, Q., Sun, Y., Jiang, Q., Du, W., Sun, C., Fu, P., and Wang, Z.: Chemical composition of aerosol particles and light extinction apportionment before and during the heating season in Beijing, China, Journal of Geophysical Research: Atmospheres, 120, 12708-12722, 10.1002/2015jd023871, 2015.

7. *Page 9, Line 23-25, please added references.*

**Reply:** We will cite more previous studies to support relevant content in the revised manuscript.

8. *Figure 7f, the date is 04/10/2016, not 04/09/2016?*

**Reply:** The date on Figure 7f will be changed to "04/09/2016".

[Figure]

Figure 6. Backward trajectories from Beijing calculated by the FLEXPART dispersion model for (a) the anthropogenic pollution case, (b) the dust-dominant case, and (c) the mixed pollution period. Variation in the standard δ value as a function of particle size for corresponding episodes: (d), (e), and (f).

9. *Page 9, Line 24-25, please added references.*

**Reply:** We will cite Liang et al. (2007), Shi et al. (2014) and Bytnerowicz et al. (2016) to support the content here.

Liang, Q., Jaeglé, L., Hudman, R. C., Turquety, S., Jacob, D. J., Avery, M. A., Browell, E. V., Sachse, G. W., Blake, D. R., Brune, W., Ren, X., Cohen, R. C., Dibb, J. E., Fried, A., Fuelberg, H., Porter, M., Heikes, B. G., Huey, G., Singh, H. B., and Wennberg, P. O.: Summertime influence of Asian pollution in the free troposphere over North America, Journal of Geophysical Research, 112, 10.1029/2006jd007919, 2007.

Shi, Y., Chen, J., Hu, D., Wang, L., Yang, X., and Wang, X.: Airborne submicron particulate (PM1) pollution in Shanghai, China: chemical variability, formation/dissociation of associated semi-volatile components and the impacts on visibility, Sci Total Environ, 473-474, 199-206, 10.1016/j.scitotenv.2013.12.024, 2014.

Bytnerowicz, A., Hsu, Y. M., Percy, K., Legge, A., Fenn, M. E., Schilling, S., Fraczek, W., and Alexander, D.: Ground-level air pollution changes during a boreal wildland mega-fire, Sci Total Environ, 572, 755-769, 10.1016/j.scitotenv.2016.07.052, 2016.

**Reply to the comments of anonymous reviewer #2 on manuscript entitled "Variability of depolarization of aerosol particles in Beijing mega city: implication in interaction between anthropogenic pollutants and mineral dust particles"**

The authors appreciate very much for reviewing our manuscript and your insight comments. As suggested, we carefully revised the manuscript thoroughly according to the valuable advices. We will response to all the comments as follows:

*The authors present a study of aerosol particles in Beijing, China, with a polarization optical particle counter (POPC). They present 8 months of statistics and 3 case studies of polluted conditions, dust conditions and a dust-pollution mixture. Especially the results presented in Fig. 7 are of interest, because they give insight in the mixing process of dust and pollution and help in the discussion about external and internal mixture. While reading the manuscript, I was missing the main new points. In the revision process, I would strongly recommend to strengthen the impact of the study: the separation of dust and pollution with a POPC and the discussion about internal or external mixture. Then, this can be applied to the statistics collected over an East Asian megacity, which lies in a global hot spot of dust and pollution mixtures.*

*The main point, why I have to reject the paper is that the retrieved depolarization values are questionable. I further checked the publication by Kobayashi et al., AE 2014, and I could not find any information on the calibration process of the depolarization ratio measured with the POPC. Especially the depolarization ratio can be influenced by multiple factors. I am sure, that the authors developed a calibration process to check the trustworthiness of the measured data, but it is missing in the publication. Also there are no comparisons of the retrieved depolarization ratios at 120° scattering angle to existing measurements or simulations of spherical or non-spherical particles. The uncertainties of the depolarization ratio values are not given. These points are essential to present trustworthy results. And as the depolarization ratio is one of the main quantities measured in this study, I have to insist on a proper calibration. Therefore, I recommend resubmitting the paper after clarifying the uncertainties of the depolarization ratio.*

**Reply:** We greatly appreciate the reviewer for insight comments on the manuscript. To respond to the reviewer's major concerns, we made thoroughly revisions and corrections according to all the insight

comments of the reviewers. Besides, more crucial information and analysis will be added in the revised manuscript, as follows:

(1) A detailed description and specification of calibration system of POPC and calibration results in this study will be added in the revised manuscript.

(2) Comparison between observation and the simulation result with T-matrix method on randomly-oriented Voronoi aggregation and elongated ellipsoid particles will be added.

(3) A comparison result of measurement of size-resolved number concentration of particles between POPC and KC52 optical particle counter will be added in the revised manuscript.

(4) Comparison of observation results between POPC measurement and LIDAR observation during several dust events in Beijing will be discussed. The cross validation on the dust events provide more direct evidence of variation in the depolarization properties of dust particles.

Besides, to make the conclusions more pertinent to this study, we revised the expression and concise the discussion about specific chemical composition and reaction. Our conclusions were useful for understanding the long-term depolarization property of aerosols in East Asia, interaction between anthropogenic pollutions and mineral dust in East Asia and their impacts on local air quality, and the sensitive relation between particle morphology and meteorological parameter. This long-term study in Beijing motivate revisit on decades of Lidar data in dust-pollution interactions, and it indicates the necessity of a reliable optical model of internally mixed polluted dust for a detailed analysis of polarization remote sensing observations.

As fas as we known, this was the first study that long-term observation on the single particle polarization properties performed in China, the aim of this study is to investigate the long-term morphology properties of particles in North China because this region has long experienced serious air pollution and dust events due to its special geographical location and industrial activities. As suggested, in the revised manuscript, we will strengthen discussion about the separation of dust and pollution and their internal or external mixture. Comprehensive study with previous studies using the polarization optical particle counter will be performed, for instance the Lidar measurement. Single particle depolarization ratio measure is able to quantitatively investigate the evolution of the mixing of dust particles during their transport. According to the depolarization ratio of their scattering signals, every mineral dust aerosols and anthropogenic pollutants can be distinguished because the direction of polarization of scattering light was identical with the incident light for spherical particle; for the non-spherical particles, a vertical polarization signal will be produced (Pan et al., 2015). Based on this characteristic, a number of studies in East Asia have been conducted that focued on the depolarization ratio of single particles. Sugimoto et al. (2015) found that backscattering depolarization ratio in polluted dust was smaller compared to pure Asia dust for the measurement at Seoul. Pan et al. (2017) reported coating processes such as heteorogeneous reaction and hygroscopic growth on the surface of dust particles play a vital role in decreasing of depolarization ratio (DR) of particles in coarse mode. There were also lots of direct evidences of internally mixed dust particle on the basis of electronic microscopy (Li and Shao, 2009; Li et al., 2011). Note that, these studies were just a case study, and long-term measurement of their mixing states and seasonal varibility in China was

still lacking. Therefore we performed an eight-month observe experiment in urban Beijing. At the present stage of this manuscript, the description about instrument calibration and uncertainty assessment was not shown, we will add relavent important information and make an detailed explanation in the revised manuscript.

[Figure]

Figure 1. A schematic diagram of the laboratory calibration process

Figure 1 shows the schematic diagram of the laboratory calibration process of POPC. The spherical polystyrene standard aerosols were Dp = 0.048 μm, 1.005 μm, 3.210 μm, 5.125 μm, 7.008 μm, 10.14 μm (JSR Life Sciences Corporation). Aerosols were generated by a nebulizer at an flow rate of 3.5 L/min,

10     desiccated by passing through a vrtically placed 45 cm Perma casing tube (MD–110–24P, GLSciences). Depolarization ratio of typical spherical particles at Dp = 5.125 μm, 7.008 μm and 10.14 μm were found to be 0.075, 0.085 and 0.102 with an pervasive uncertainty of ± 0.01,and was almost zero for the fine particles ( Dp = 0.048 μm, 1.005 μm ) (Figure 2). In this study, the DR values of aerosols in coarse mode were found to be centered on ~ 0.3, much larger than calibration results, indicate that coarse mode aerosol

15     particles at the site was non-spherical generally.

[Figure]

Figure 2. The calibration results of POPC. Optical size of the particle converted from a forward scattering signal at 60 degree (left); Depolarization of the standard particles detected from a backward scattering signal at 120 degree (right).

Due to the limitation of standard aerosols, we only did the calibration of the spherical particles. For non-
5   spherical aerosols, we did some simulation to show the depolarization ratio at a scattering angle of 120 °. The polarization property of randomly oriented elongated ellipsoid particles was simulated on the basis of the T-matrix methodology (Dubovik et al., 2006). The non-sphericity is indicated by the aspect ratio (the ratio of long axis to its orthogonal short axis). As indicated in Figure 3, the observed depolarization ratio ( range from 0.1-0.5, and centered on 0.3 ) in this study corresponded to an aspect ratio of 1.20 -
10  1.55 for coarse mode dust particles, with mode value 1.50. During the dust-dominant period on April 10, the aspect ratios of the dust particles were estimated to be 1.51 as the depolarization ratio of the dust particles was 0.34. While on polluted dust period on March 4, the aspect ratios of the dust particles were estimated to be 1.48 as the depolarization ratio of the dust particles was 0.30, providing that the dust particles has a great possibility to underwent partial deliquescent and hygroscopic growth on the surface
15  in mixed pollution period.

[Figure]

Figure 3. Theoretical simulation of the depolarization ratio of randomly oriented elongated ellipsoid particles as a function of the aspect ratio at fine mode: Dp = 0.5 μm,1 μm, and Dp = 5 μm, 7μm and on the basis of the T-matrix methodology.

20  Such characteristic was also well predicted by optical model considering particles of Voronoi aggregation (Ishimoto et al., 2010) (Figure 4). Theoretical simulation showed that an change in refractive index could affect the DR value evidently, for the coarse modal particles observed in this observation experiment, the simulation results showed that the depolarization ratio showed a leveling off tendency at $0.31 \pm 0.02$, confirmed with the depolarization ratio observed in our study.

[Figure]

Figure 4. Theoretical calculation of depolarization ratio (at 120 backward direction) as a function of particle size for different refractive index.

As the reviewer suggested, providing the comparison that between the retrieved depolarization ratios at 120◦ scattering angle to existing measurements would better enhance the credibility of our data. The reality is that we are the first study that focused on the single particle polarized optical spectrometer in China. As far as I know, Lidar measurement and satellite on-board remote sensing (CALIPSO) (Glen et al., 2013) could provide tempo-spatial profile of volumn integrated depolarizaiton ratio, however the defination of Lidar's depolarizaiton ratio (S/P, backward at angle of 180º) is different from POPC and the two methods all persuming externally mixed of spherical particles and dust aerosols. Beyond that, there is no relevant comercial instrument that can make such measurement. Mass concentration of PM was reconstructed on the basis of number concentration of particles measured by POPC and particles density. The particle density was assumed to increase linearly from 1.77 g/cm$^3$ (0.5 µm) to 2.2 g/cm$^3$ (10 µm). The result compared well with commercial optical particle counter (KC52, RION, as shown in Figure 5), especially in the coarse mode size range observation. We will add relevant literatures and additional remarks in the context.

[Figure]

Figure 5. The comparison of number concentrations measured by POPC and OPC (KC52, RION).

The uncertainties of the depolarization ratio values was affected by various factors, including voltage variance of power supply ($\sigma_{vol}^2$), emvironment water content ($\sigma_{WC}^2$) and complex refraction index ($\sigma_{nf}^2$) of the aerosol. Light scattering signals are stored in the form of electrical impulses, the voltage instability will affect the pulse signals. The power supply voltage of POPC was set 15 V, and have a fluctuation by 5% on test. The residence time of diluted air mass in POPC was estimated to be 0.7 s, which was generally sufcient for aerosol particles to achieve equilibrium before measurement in the detecting chamber. It suggested that the wet particles tend to shrink due to loss of water. Zhang et al. (2015) reported aerosol backscattering coefficient increased 25% as the RH increased from 40% to 85 %. The measurement uncertainty in DR was estimated to be 10% caused by humidy change. Theoretical simulation also indicated that an increase in the imaginary part of the refractive index could reduce the DR value evidently (Ishimoto et al., 2010), according to the simulation experiments variations in the particle's refractive index (the real and imaginary part) can explain 6% depolarization variability (Figure 4). Under the calculation method ( $\sigma_{DR} = \sqrt{\sigma_{vol}^2 + \sigma_{nf}^2 + \sigma_{WC}^2}$ ), we estimate the uncertainty of depolarization ratio was < 13%. As indicate in Figure 14 in the manuscript, the depolarization ratio has a decrease of 50% and 43% for 3 μm and 5 μm aerosols, we conclude that dust particles in the atmosphere does contain hygroscopic component.

Pan, X., Uno, I., Hara, Y., Kuribayashi, M., Kobayashi, H., Sugimoto, N., Yamamoto, S., Shimohara, T., and Wang, Z.: Observation of the simultaneous transport of Asian mineral dust aerosols with anthropogenic pollutants using a POPC during a long-lasting dust event in late spring 2014, Geophysical Research Letters, 42, 1593-1598, 10.1002/2014gl062491, 2015.

Sugimoto, N., Nishizawa, T., Shimizu, A., Matsui, I., and Kobayashi, H.: Detection of internally mixed Asian dust with air pollution aerosols using a polarization optical particle counter and a polarization-sensitive two-wavelength lidar, Journal of Quantitative Spectroscopy & Radiative Transfer, 150, 107-113, 10.1016/j.jqsrt.2014.08.003, 2015.

Pan, X., Uno, I., Wang, Z., Nishizawa, T., Sugimoto, N., Yamamoto, S., Kobayashi, H., Sun, Y., Fu, P., Tang, X., and Wang, Z.: Real-time observational evidence of changing Asian dust morphology with the mixing of heavy anthropogenic pollution, Sci Rep, 7, 335, 10.1038/s41598-017-00444-w, 2017.

Li, W. J., and Shao, L. Y.: Observation of nitrate coatings on atmospheric mineral dust particles, Atmos Chem Phys, 9, 1863-1871, DOI 10.5194/acp-9-1863-2009, 2009.

Li, W. J., Zhang, D. Z., Shao, L. Y., Zhou, S. Z., and Wang, W. X.: Individual particle analysis of aerosols collected under haze and non-haze conditions at a high-elevation mountain site in the North China plain, Atmos Chem Phys, 11, 11733-11744, 10.5194/acp-11-11733-2011, 2011.

Dubovik, O., Sinyuk, A., Lapyonok, T., Holben, B. N., Mishchenko, M., Yang, P., Eck, T. F., Volten, H., Munoz, O., Veihelmann, B., van der Zande, W. J., Leon, J. F., Sorokin, M., and Slutsker, I.: Application of spheroid models to account for aerosol particle nonsphericity in remote sensing of desert dust, Journal of Geophysical Research-Atmospheres, 111, 10.1029/2005jd006619, 2006.

Ishimoto, H., Zaizen, Y., Uchiyama, A., Masuda, K., and Mano, Y.: Shape modeling of mineral dust particles for light-scattering calculations using the spatial Poisson–Voronoi tessellation, Journal of Quantitative Spectroscopy and Radiative Transfer, 111, 2434-2443, 10.1016/j.jqsrt.2010.06.018, 2010.

Glen, A., and Brooks, S. D.: A new method for measuring optical scattering properties of atmospherically relevant dusts using the Cloud and Aerosol Spectrometer with Polarization (CASPOL), Atmos Chem Phys, 13, 1345-1356, 10.5194/acp-13-1345-2013, 2013.

Zhang, L., Sun, J. Y., Shen, X. J., Zhang, Y. M., Che, H., Ma, Q. L., Zhang, Y. W., Zhang, X. Y., and Ogren, J. A.: Observations of relative humidity effects on aerosol light scattering in the Yangtze River Delta of China, Atmos Chem Phys, 15, 8439-8454, 10.5194/acp-15-8439-2015, 2015.

*Further comments for improving the manuscript:*

1. *Add an outline of the paper at the end of the introduction*

   **Reply:** As the reviewer suggested, we will add an outline of the paper at the last paragraph of the introduction as follows:

   "In this study, a comprehensive ground-based measurement of depolarization properties of aerosol particles was performed at an urban site in Beijing, at the State Key Laboratory of Atmospheric Boundary Layer Physics and Atmospheric Chemistry (LAPC, Longitude: 116.37E; Latitude: 39.97N), Institute of Atmospheric Physics/Chinese Academy of Sciences. Morphological variability of ambient aerosol particles were investigated from November 2015 to July 2016 using a Polarization Optical Particle Counter (POPC), and the seasonal characteristics of depolarization ratio (δ) of atmospheric aerosols were explained. Three pollution events including anthropogenic pollutants, mineral dust-dominant and polluted dust were classified according to its size distribution and δ value, as well as trajectory analysis, to investigate the interactions between dust particles and pollutants. The objective of this study focuses on the variation of the δ value of aerosol particles and its relationship with secondary pollutants. This study represents the first time such long-term observation of ambient aerosol morphology performed in the megacity of Beijing, and provides more applicable data for evaluating the mixing processes of atmospheric aerosols and their climate impact."

2. *Mention, which size range is covered with your optical particle counter.*

   **Reply:** The effective size range of the optical particle counter was 0.5-10 μm, we will add it into the manuscript.

3. *The term "fine mode particles" normally refers to particles with a diameter smaller than 1 μm. PM1 data would be helpful to assess the fine mode particles.*

   **Reply:** Thanks for the reviewer's suggestion, we carefully checked relevant studies on atmospheric aerosols, and find that fine mode particles usually refers to particles with an aerodynamic diameter below 2.5 microns (Adams et al., 2001; Rathnayake et al., 2017a; Rathnayake et al., 2017b) and $PM_1$ (Bari et al., 2015; Shi et al., 2014) was often referred to as the sub-micron scale.

   Adams, H. S., Nieuwenhuijsen, M. J., and Colvile, R. N.: Determinants of fine particle (PM2.5) personal exposure levels in transport microenvironments, London, UK, Atmospheric Environment, 35, 4557-4566, 10.1016/s1352-2310(01)00194-7, 2001.

Rathnayake, C. M., Metwali, N., Jayarathne, T., Kettler, J., Huang, Y., Thorne, P. S., amp, apos, Shaughnessy, P. T., and Stone, E. A.: Influence of rain on the abundance of bioaerosols in fine and coarse particles, Atmos Chem Phys, 17, 2459-2475, 10.5194/acp-17-2459-2017, 2017a.

Rathnayake, C. M., Metwali, N., Jayarathne, T., Kettler, J., Huang, Y. F., Thorne, P. S., O'Shaughnessy, P. T., and Stone, E. A.: Influence of rain on the abundance of bioaerosols in fine and coarse particles, Atmos Chem Phys, 17, 2459-2475, 10.5194/acp-17-2459-2017, 2017b.

Bari, M. A., Kindzierski, W. B., Wallace, L. A., Wheeler, A. J., MacNeill, M., and Heroux, M. E.: Indoor and Outdoor Levels and Sources of Submicron Particles (PM1) at Homes in Edmonton, Canada, Environmental Science & Technology, 49, 6419-6429, 10.1021/acs.est.5b01173, 2015.

Shi, Y., Chen, J., Hu, D., Wang, L., Yang, X., and Wang, X.: Airborne submicron particulate (PM1) pollution in Shanghai, China: chemical variability, formation/dissociation of associated semi-volatile components and the impacts on visibility, Sci Total Environ, 473-474, 199-206, 10.1016/j.scitotenv.2013.12.024, 2014.

4. *State clearer the difference between the depolarization ratio used in lidar studies (s-polarized to p-polarized ratio at 180◦ backscatter) and the depolarization ratio used in the POPC study (s-polarized to s+p polarized ratio at 120◦ backscatter). To add value to the lidar studies by using single particle analyses, it would be better to use the same depolarization ratio or at least to present a method to convert the POPC depolarization ratio to a lidar retrieved depolarization ratio. But I dare, that this would be rather complicated for non-spherical particles.*

**Reply:** The main difference between the depolarization ratio ($\delta a$) used in LIDAR and the POPC was: Lidar measured was all the particles' depolarization ratio in backward scattering angle 180° in a beam volume. The overall depolarization ratio of dust particles might be underestimated due to substantially presence of small spherical particles in the dust plume. It means external mixing of a large amount of fine particles with mineral dust aerosols also results in a lower $\delta a$; POPC measured all the single particle's depolarization ratio in the backward scattering without interference. According to theoretical simulation and optical lens design, POPC only measure depolarization ratio of scatter light at angle of 120°. Considering that the scattering phase function of particles was not linearly varying at different angles, especially for non-spherical particles, depolarization ratio at backscatter 120 ° was difficult to switch to the results of 180 °. Now we are working on a project of remould the light road of POPC, in the new version, we design to detect the scattered light information at around 180 °. In our next research, we would like to follow the review's suggestion to try provide a comparison for LIDAR result.

5. *Add the distance and the direction (north, east,…) to the description of the two additional measurements sites (Olympic Sport Centre state control site and meteorological station) and not just the coordinates. It makes it easier for the reader.*

**Reply:** Thanks to the reviewers' suggestions, we will add information about the distance and the direction to the description of the two additional measurements sites.

[revised manuscript text omitted]